# Generating a Diversity of Challenging Programming Puzzles with Autotelic Generative Models

**Julien Pourcel,**
Inria

**Cédric Colas**
MIT, Inria

**Gaia Molinaro**
University of California, Berkeley

**Pierre-Yves Oudeyer**
Inria

**Laetitia Teodorescu**
Inria

## Abstract

The ability to invent novel and interesting problems is a remarkable feature of human intelligence that drives innovation, art, and science. We propose a method that aims to automate this process by harnessing the power of state-of-the-art generative models to produce a diversity of challenging yet solvable problems, here in the context of Python programming puzzles. Inspired by the intrinsically motivated literature, Autotelic CodE Search (ACES) jointly optimizes for the diversity and difficulty of generated problems. We represent problems in a space of LLM-generated semantic descriptors describing the programming skills required to solve them (e.g. string manipulation, dynamic programming, etc.) and measure their difficulty empirically as a linearly decreasing function of the success rate of *Llama-3-70B*, a state-of-the-art LLM problem solver. ACES iteratively prompts a large language model to generate difficult problems achieving a diversity of target semantic descriptors (goal-directed exploration) using previously generated problems as in-context examples. ACES generates problems that are more diverse and more challenging than problems produced by baseline methods and three times more challenging than problems found in existing Python programming benchmarks on average across 11 state-of-the-art code LLMs.

## 1 Introduction

Humans are not only talented problem solvers, they are first of all remarkable *problem generators* — generating endless streams of new problems for themselves and others Chu and Schulz [2020], Molinaro and Collins [2023]. We set build problems for others to learn, set challenges to ourselves, [Burton and Hiron, 2008], aggregate problems to train and test AI models [Hendrycks et al., 2020, Chen et al., 2021], and come up with new problems that drive innovation in art and science [Gromov, 2018, Chu et al., 2024]. This intrinsic drive to generate problems for oneself — the *autotelic property* — has further been argued to drive the capacity for adaptation and open-ended learning in both human [Chu and Schulz, 2020] and machines [Schmidhuber, 2013, Herrmann et al., 2022, Colas et al., 2022]. Automating this problem-generation process would have numerous positive applications: for instance, designing exercises tailored to optimize the learning experience of every human or machine learner (automatic curriculum learning Portelas et al. [2020]); or facilitating the generation of evaluation protocols (human tests, machine learning benchmarks). It would provide the necessary curriculum for open-ended learning machines [Colas et al., 2022] and may be a key component of automated scientific discovery [Grizou et al., 2020, Etcheverry, 2023].

We propose to leverage machine learning — a set of tools usually targeted at *solving* problems — to automate the *generation of a diverse set of interesting problems*, here in the domain of Python programming puzzles. Programming puzzles indeed represent an open-ended space of problems to

38th Conference on Neural Information Processing Systems (NeurIPS 2024).

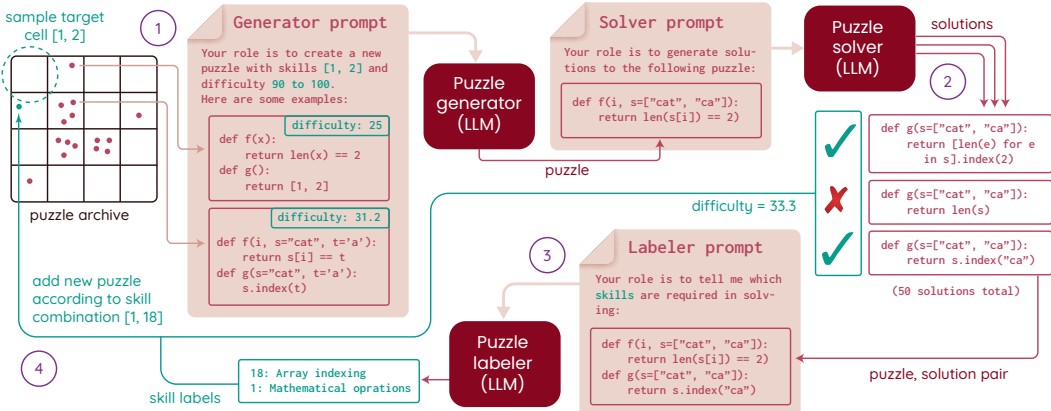

Figure 1: **Overview of the ACES algorithm.** ACES iteratively generates a diverse set of challenging programming puzzles. First, a target *cell* corresponding to a combination of programming skills is sampled from a puzzle archive (1), and puzzles from filled neighboring cells — prioritized by difficulty — are selected as examples and given to a puzzle generating LLM. It generates a new puzzle with the desired skill combination that an LLM solver tries to solve 50 times (2). If never solved, the puzzle is discarded. An LLM describes the skills needed to solve the puzzle (3) and the puzzle, along with its computed difficulty score, is added to the puzzle archive.

explore: from simple string manipulations to complex dynamic programming or open mathematical puzzles [Schuster et al., 2021]. We qualify problems as *interesting* when they are challenging yet solvable. This can be estimated by computing the empirical difficulty of a puzzle for a particular solver: out of 50 attempts, the solver should solve the problem at least once (solvability) but as rarely as possible (difficulty). In contrast with natural language instruction domains, our puzzle domain lets us objectively check the validity of a given solution by simply running a Python interpreter. This domain thus affords both an open-ended space to explore (diversity search) and an objective quality measure to maximize.

The standard approach for problem generation simply queries pretrained generative models with few-shot examples or specific instructions [Haluptzok et al., 2022, Honovich et al., 2023, Gunasekar et al., 2023]. This amounts to sampling from a stationary distribution such that the quality and novelty of generated problems reflect those of the problems found in the training data. Instead, we introduce *Autotelic CodE Search* (ACES), an *autotelic generative model* that steers pretrained generative models to produce a diversity of challenging puzzles by iteratively reusing previously generated puzzles as examples to guide the production of *newer and harder problems*.

ACES builds on *Map-Elites* [Mouret and Clune, 2015], an evolutionary quality-diversity (QD) method [Pugh et al., 2016]. In all QD algorithms, the user first needs to define a *descriptor function* mapping each generated outcome (here, each problem) to a numerical representation that will be used to measure diversity. Programming puzzles are high-dimensional objects that we chose to represent by the *set of programming skills required to solve them* — a high-level semantic description space that better captures intuitive notions of puzzle diversity than pretrained embedding representations. We obtain these by asking a large language model (LLM) to label each problem a set of skills from a list of 20 possible ones. Just like Map-Elites, ACES maintains an archive of generated outcomes grouped by skill sets (descriptor niches) and optimizes quality (here difficulty) locally within each niche. While Map-Elites randomly mutates solutions sampled from the archive in hopes of discovering a new niche or finding higher-performing solutions, ACES performs an *explicit goal-directed exploration* [Colas et al., 2022]. At each iteration, ACES targets a goal descriptor, carefully selects relevant and challenging example problems from the archive and, conditioned on these, prompts an LLM to generate a more difficult problem-solution pair achieving the goal. The new puzzle is labelled, evaluated and, if valid, is added to its corresponding descriptor niche in the archive. Across iterations, the archive gets filled with more diverse and challenging puzzles which provide higher quality examples to guide the generation of yet more diverse and challenging puzzles (see Figure 1).

We show that ACES generates a wider diversity of more challenging problems than both standard generative approaches [Haluptzok et al., 2022, Honovich et al., 2023] and existing algorithms based on Map-Elites [Bradley et al., 2023a]. Finally, we show that generated problems are harder to solve

than those found in existing Python programming benchmarks, this for all the 11 state-of-the-art LLM-based problem solvers we tested. Whereas the HumanEval+ benchmark is starting to saturate (*GPT-4-turbo* achieves a pass@1 of 86.6%, *CodeQwen1.5-7B-Chat* 78.7%, [Liu et al., 2024])[1], the best solver (*Mixtral-8x22B-Instruct-v0.1*) only achieves a pass@1 of 47.3% on our problems. This work paves the way for automating the design of harder benchmarks whose difficulty is calibrated using LLM solvers themselves, eventually allowing evaluations to increase in difficulty as models improve.

## 2 Related Work

**Open-ended exploration algorithms** Our puzzle-generating method is at the intersection of two research lines: evolutionary computing [Lehman and Stanley, 2011a,b, Mouret and Clune, 2015, Pugh et al., 2016, Cully and Demiris, 2018a, Lehman et al., 2022] and intrinsically-motivated learning [Baranes and Oudeyer, 2013, Etcheverry et al., 2020, Forestier et al., 2022, Colas et al., 2022]. Beginning with *novelty search* [Lehman and Stanley, 2011a,c], the evolutionary approach to exploration expanded with the invention of quality-diversity algorithms [QD: Lehman and Stanley, 2011b, Mouret and Clune, 2015, Cully and Demiris, 2018a], a set of methods striving to evolve a diverse population of locally-performant solutions via the undirected mutation of existing solutions. A parallel line of research introduced *goal-directed* exploration processes, also called *autotelic learning*, where agents learn to represent and sample their own goal as a way to direct the diversity search [Baranes and Oudeyer, 2013, Forestier et al., 2022, Colas et al., 2022]. Although autotelic methods were first developed to model the open-ended development of children in skill learning robots Baranes and Oudeyer [2013], Moulin-Frier et al. [2014], Oudeyer and Smith [2016], they also proved effective in the automatic exploration of complex systems, either simulated [Reinke et al., 2019, Etcheverry et al., 2020] or physical [Grizou et al., 2020].

**LLM-augmented exploration** LLMs can be useful in various parts of quality-diversity and exploration algorithms. Their capacity to generate appropriate variations of existing text has been used to implement mutation [ELM: Lehman et al., 2022] and crossover [Meyerson et al., 2023] operators within QD, with applications to neural architecture search [Chen et al., 2023, Nasir et al., 2023]. Following the recent trend in learning from AI feedback [Bai et al., 2022, Lee et al., 2023] and using LLM-as-judge methods [Zheng et al., 2024], recent work has used LLM responses as models of *interestingness* [Zhang et al., 2023, Klissarov et al., 2023, Sachdeva et al., 2024] within RL, while others have augmented Map-Elites with LLM-based quality judgments and semantic descriptors for creative writing [Bradley et al., 2023b] or adversarial prompt generation [Samvelyan et al., 2024]. These last two are close to our ELM baseline, but in contrast to these works we use a grounded empirical difficulty metric as quality compared to AI feedback which is less grounded and might be innacurate. ACES additionally uses LLM-augmented goal generation, which echoes recent methods leveraging generative models for open-ended goal-based exploration [Colas et al., 2023, Wang et al., 2023a, Du et al., 2023]. In contrast to these works we optimize for both the difficulty and the diversity of the generated puzzles.

**LLMs for generating code and instruction datasets** This work is also linked to prior approaches for generating synthetic code and instruction data, mostly in data augmentation contexts. Seminal works leveraged a standard generative approach by prompting an LLM to generate new problems using as few-shot examples problems sampled from an existing dataset [Haluptzok et al., 2022, Honovich et al., 2023, Roziere et al., 2023], or problems generated at previous iterations [Wang et al., 2023b]. Closer to the goal-targeting of ACES, Eldan and Li [2023] generates diverse training data by asking an LLM to write stories employing a combinations of words randomly sampled from a large list. Gunasekar et al. [2023], Abdin et al. [2024] build upon this approach to generate programming textbooks by mixing subjects and audiences, and generates exercises by randomizing the exercise name. *Evol-Instruct* is an evolutionary method that iteratively generates language or code instructions by applying prompts that modify previously generated problems by, among other things, increasing their difficulty [Xu et al., 2023, Luo et al., 2023]. They do not optimize for diversity and do not use actual difficulty measurements, relying on the LLM's problem modification to increase it. Finally, *Skill-Mix* [Yu et al., 2023] generates language evaluations for LLMs by generating problems involving combinations of skills in the language domain and grading models using GPT-4; they do not use

---

[1] https://evalplus.github.io/leaderboard.html

```python
def f(ls: List[str]):
    """Divide the decimal representation of 8^88 up into strings of
        length eight."""
    return "".join(ls)==str(8**88) and all(len(s)==8 for s in ls)
def g():
    return [str(8**88)[i:i+8] for i in range(0,80,8)]
assert f(g()) == True
```

Figure 2: **Puzzle example.** A simple programming puzzle and its solution from the P3 dataset [Schuster et al., 2021]. A solution function `g` must return a valid solution such that `f(g()) == True`.

any few-shot examples, do not optimize nor measure difficulty for an LLM while generating their problems.

## 3 Methods

### 3.1 Programming puzzles and the P3 dataset

The *Python Programming Puzzles dataset* (P3) contains 1715 puzzle-solution pairs where each puzzle is defined by a short test program `f` designed to verify the validity of solution programs `g` such that valid solutions satisfy `f(g()) == True` when run in an interpreter, see example in Figure 2 [Schuster et al., 2021]. P3 puzzles span problems of various difficulties that involve different programming skills: from classic problems (tower of Hanoi) and string manipulations, to factoring problems, dynamic programming, or even open problems in computer science or mathematics. The P3 dataset is split into training and testing datasets ($N = 636$ and 1079 respectively). Both datasets are pre-filtered to examples shorter than 1024 tokens to accommodate for limited context windows in LLMs.

### 3.2 Generating diverse sets of challenging puzzles

In this work, we aim to generate sets of puzzles that are collectively as diverse and on average as difficult as possible. In this section, we first define the difficulty metric we use, and then how we quantify diversity.

**Empirical puzzle difficulty** We measure the empirical difficulty of a puzzle with respect to a target LLM solver as the opposite of the solver's competence on that puzzle. We measure competence using the standard pass@k metric with k=1: the number of valid solutions generated after k=1 attempts (which is simply the success rate) [Chen et al., 2021]. We estimate the pass@1 competence over $N = 50$ solving attempts and report the empirical difficulty (puzzle fitness $\mathcal{F}$) as its negative success rate rescaled to the 0–100 range:

$$\mathcal{F}(\texttt{f}) = \begin{cases} (-\text{pass@1}(\texttt{f}, \text{LLM}) + 1) \times 100 & \text{if pass@1} \neq 0 \\ -\infty & \text{otherwise} \end{cases} \quad (1)$$

The more difficult the puzzle is, the higher its fitness. Puzzles for which no solution is found are considered invalid and are discarded. The prompt used for generating solutions can be found in the Appendix. Compared to LLM-based methods of assessing the quality of a sample (typically, critic or LLM-as-judge methods), this difficulty-based metric measures a ground-truth objective: how hard a given puzzle is for a target LLM solver. Our experiments found that this difficulty measure mostly transfers across models: a puzzle that is harder for one model is often harder for others (see Section 4.5). Although our difficulty metric is rather expensive to compute, it captures exactly the intended objective and is thus harder to hack or overfit compared to objectives based on LLM feedback [Zheng et al., 2024, Sachdeva et al., 2024]. This said, an existing possibility for hacking the difficulty metric is to import a random number generator library and to make the `f` function return True with probability 1/50. We have not observed this phenomenon in our experiments.

**Skill combination diversity** The diversity measure we choose to optimize will have an important impact on the distribution of generated puzzles. Ideally, the set of puzzles should be diverse in their structures and topics. We thus define a set of 20 tags corresponding to different programming skills

and label each programming puzzle with the combination of skills needed to solve it (reminiscent of Skill-Mix [Yu et al., 2023]). We then measure *semantic diversity* as the number of unique skill combinations for which there is at least one representative in our generated set. We refer to a tag combination as a *niche* or a *cell* in the rest of the paper. We sample our 20 tags from a larger list generated by an LLM (*GPT-4-0125*) and validated against lists of programming topics covered in classic computer science textbooks and competitive programming platforms (LeetCode and HackerRank). The complete list of tags is given in Appendix Section! A.2 and contains items such as *Recursion*, *Geometry and coordinate problems* or *Hashing*. We limit combinations to a maximum of 5 skills out of 20 possible skills, which gives us 21,700 total niche combinations. This helps avoid unrealistic skill combinations. We prompt an LLM to label puzzles with a set of programming skills as in Bradley et al. [2023b], Samvelyan et al. [2024].

**Embedding diversity**   The diversity measure is based on an LLM's feedback and is thus subject to inaccuracies, especially if this metric is used as an optimization target. As complemetary measures of diversity, we propose *embedding diversity* metrics that estimate diversity in a variety of pretrained embedding spaces that were not used by the algorithms. This measure is computed as the average pairwise cosine distance between the embeddings of all problems generated in the set.

### 3.3   ACES: Autotelic CodE Search

In this section we present ACES, an exploration method that samples target niches (sets of descriptors) as goals and prompts an LLM with relevant challenging puzzles sampled from an archive of previously generated puzzles to reach them. Like ELM, which we use as baseline, ACES uses an LLM to generate mutations of existing samples from the archive. Compared with ELM, where the LLM is instructed to produce a variation of a given puzzle without a specific objective in mind, ACES instructs the LLM to generate a new puzzle based on its examples. The generated puzzles are then evaluated for fitness and labeled, and added to the archive before the next generation round. The cells are initialized with the deduplicated[2] P3 train set puzzles. An overview of the algorithm is given in Figure 1.

**Sampling a goal and relevant examples**   First, we sample a cell uniformly, then we look at the 3 closest cells for which there is at least one puzzle (the target cell can be one of them). For each of these neighbor cells, we select one puzzle and add it as an example to the prompt. The prompt (available in the Appendix) instructs the LLM to produce puzzles in the target cell (corresponding to a combination of skills) using the 3 sampled puzzles as examples. To select examples from one cell, we normalize the range of fitness scores in this cell between 0 and 1 and then sample from a softmax distribution from these normalized fitnesses with a temperature of 0.2. To guide the model towards generating harder puzzles, we added the difficulty score $\mathcal{D}$ of each puzzle in the prompt (ranging from 0 to 100), instructing the model to reach a score between 90 and 100 when generating new puzzles. The intuitions underlying the design of ACES are: to drive discovery of puzzles with novel skill combinations, we rely on the LLM recombining elements from puzzles close to the target cell along with very often selecting target cells without any representatives. To drive discovery of harder puzzles, we rely on sampling harder puzzles more often as examples and the assumption that harder examples, along with the explicit instruction to create hard puzzles, will lead to more difficult generated puzzles.

**Generator and labeler LLMs**   Once the prompt is built, the generator LLM is instructed to produce five new puzzles following the desired instruction. For each of these generated puzzles, their fitness is computed as the negative success rate over 50 attempts (Equation 1). If the puzzle is not solved by the solver model in 50 attempts, it is considered unsolvable and discarded. For each solvable puzzle, a solution is sampled randomly from the valid ones, and the (puzzle, solution) pair is then described by an LLM. This description is a short text explaining what the puzzle is (like docstrings in P3, or instructions in HumanEval). The (puzzle, solution, description) triplet is then handed to a labeler LLM that produces the skill tags for the puzzle. The description and labeling prompts can be found in Appendix.

---

[2]Every P3 puzzle comes in several instances with different arguments. We randomly sample one instance per puzzle, which gives use 155 seed puzzle instances.

**Avoiding label hacking**   The reason we generate the puzzle description separately from the puzzle itself is that in preliminary experiments, we observed a form of *label hacking* where the generator LLM, when tasked with generating puzzles with a particular combination of skills, generated generic simple puzzles and listed the relevant skills in the description even if they were not required in solving the puzzle. These puzzles were then wrongly tagged with a rare combination of skills and were thus oversampled as examples for the next generations, leading to the propagation of misleading descriptions. Describing the puzzle independently of all references to desired programming skills mitigates this sort of label hacking. It needs to be noted that all forms of AI feedback, be it for the fitness or for the tags, are susceptible to hacking; all methods relying on the optimization of AI feedback signals should consider and mitigate this form of hacking.

## 3.4   Baselines and variations

**ELM with semantic categories**   We test an ablation of goal-directedness to study its impact, and this is exactly applying ELM to our task by using the same quality and descriptor function. To create new individuals, a cell with at least one representative is sampled at random, an individual is sampled in this cell using the same quality-based sampling mechanism as ACES, and an LLM is instructed to compose a variant of this puzzle using two other random puzzles from the overall archive as few-shot examples of what the puzzle domain looks like. The few-shot examples are given so ELM and ACES both use the same number of examples in their prompts. We term this baseline ELM (in reference to Evolution through Large Models which inspired this work), even if the original ELM did not use any form of AI feedback. We study an additional method combining ACES and ELM. This method also mutates a single puzzle, but the LLM is instructed to produce a targeted variation by trying to reach a target cell like in ACES. We term this method ACES-ELM.

**ELM with CVT + embeddings**   We additionally study the impact of using natural language-based descriptors to measure and optimize for diversity, compared with embeddings. Previous QD methods [Vassiliades et al., 2017] aiming to extend their descriptor spaces to larger dimensions used centroidal Voronoi tessellations [CVT: Du et al., 1999] to partition the space into a tractable number of cells. In the ELM-CVT baseline, we use the P3 train puzzle embeddings as seeds to generate 40000 points in embedding space by adding Gaussian noise with mean 0 and standard deviation 0.12. We then cluster all these points into 10000 clusters which are used as centroids for our Voronoi cells. ELM-CVT behaves as ELM, but the cells are defined with their Voronoi cells instead of programming skill combinations. Embeddings are computed with the *code5p-embedding* model.

**Static Gen**   Our final baseline is a standard generative method using a static few-shot prompting mechanism for puzzle generation, similar to Unnatural Instructions [Honovich et al., 2023] and Haluptzok et al. [2022]. At each time step, 3 puzzles are randomly sampled from the P3 train set and given as examples to an LLM for generating new puzzles. Generated puzzles are not re-used as examples. Puzzle generation prompts for all methods are available in the Appendix.

# 4   Results

## 4.1   Experimental details

Puzzle generation, solution generation, description generation, and puzzle labeling are all implemented with the state-of-the-art open source model *Llama 3 70B*, quantized in 4 bits, with a temperature parameter of 0.8.[3] We repeat all experiments using 3 different random seeds and report the means and standard deviations of the results. Each experiment was performed on 1 node of 4 Nvidia Tesla V100 SXM2 32 GB, with 160 GB RAM, for about 20 hours using the vLLM library [Kwon et al., 2023]. Each experiment is run for 40 generations, where each generation corresponds to 160 puzzles generated by the puzzle generator — a total of 6400 puzzle generation attempts per run.

---

[3]Model available at:
https://huggingface.co/TechxGenus/Meta-Llama-3-70B-Instruct-GPTQ

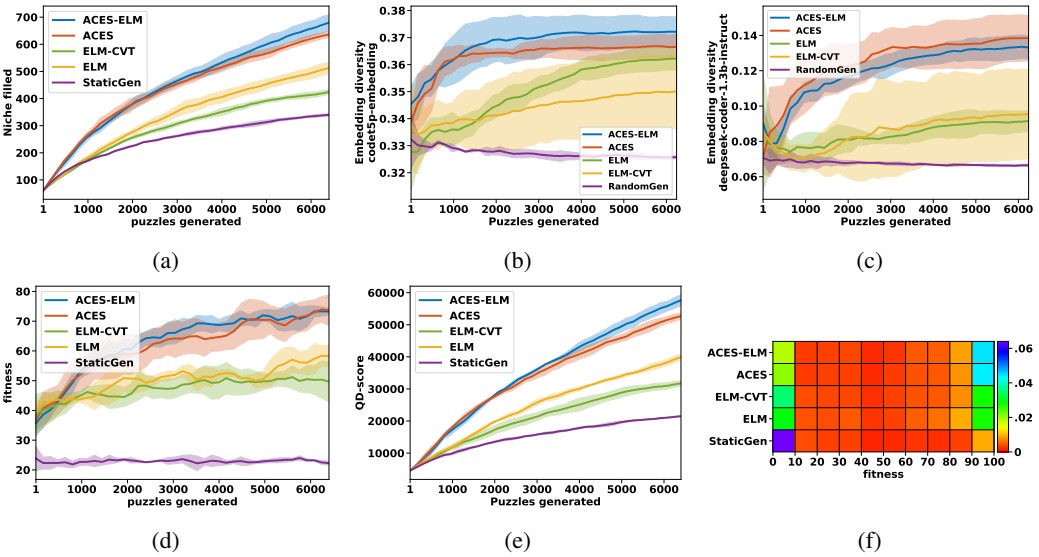

Figure 3: **ACES generates more diverse and more difficult problems.** *Diversity* (first row): semantic diversity (a), embedding diversity with the *codet5p* model (b) and the *deepseek-coder-1.3b* model (c). *Fitness* (second row): average fitness of valid puzzle generated over the last 160 generation attempts (d), QD-score (e) and distributions of fitness values over whole archives (f). ACES variants outperform baselines in terms of diversity, fitness and QD score (aggregated measure).

## 4.2 Quality of LLM-generated skill labels

We represent generated problems with skill descriptors generated by an LLM. This allows us to characterize abstract, semantic aspects of the generated problems that would be hard to capture with hand-written descriptor functions, and thus lets us optimize diversity in a space that is more aligned with human intuitive notions of variations in this programming domain. However, LLM labeling is stochastic and can be mis-aligned with human judgements. To validate the labeling process, we tagged 60 puzzles with semantic descriptors selected from the list of 20 (see Section 3.2) and, using them as ground truth, report a precision of 0.71, a recall of 0.75 and an F1 score of 0.73. Qualitatively, we found the labeler to be generally competent, always detecting the most salient skill descriptor and almost never labeling a puzzle with a totally irrelevant descriptor — although it did sometimes answer quite literally, calling a problem *set problem* when it used Python's *set* function. Appendix Figure 7 shows that the labeler uses almost all descriptors across the experiments, generally using 2-3 descriptor labels per puzzle.

## 4.3 ACES generates more diverse and challenging problems than existing approaches

**More diverse** The ability of an algorithm to generate diversity can be measured as the number of descriptor niches it manages to fill — stronger algorithms can reach more niches. On this metric, ACES and its variant ACES-ELM vastly outperform other baselines, reaching up to 700 different niches by the end of the experiment (Figure 3a). This shows that our approach can effectively generate a diversity of problems with respect to a target description space provided by the user: here the set of 20 programming skills defined in Section 3.2. Although ELM also leverages the same semantic descriptors, its undirected mutation operator does not seem to optimize diversity as effectively as the goal-directed generation operator of ACES variants.

However, the imperfection of the skill labeling could cast doubts on this result: could it be that ACES variants learn to generate problems that hack the labeler and force it to generate diverse labels that do not capture the true properties of the problems, this despite our preventive measures (see Section 3.3)? Furthermore, ELM-CVT and StaticGen do not leverage the semantic representation space, which may explain why they generate lower diversity in that space. To validate our results, we also measure *embedding diversity* metrics in a variety of other sentence embedding spaces not used at any point during training. This measure is obtained by embedding all generated puzzles with

a given model, then computing the average cosine distance between all pairs of embeddings in the set. Figures 6a and 3c show these metrics for two standard embedding models: *codet5p-embedding* and *deepseek-coder-1.3b*: ACES and its variant significantly outperform others here too. ELM-CVT represents problems and optimizes for diversity in the *codet5p* embedding space, which explains its ability to generate higher diversity in that space than ELM (Figure 6a). However, this diversity does not transfer well to other diversity metrics (Figure 3a and Figure 3c), while the semantic diversity optimized by ACES variants does transfer well across diversity metrics (Figures 3a to 3c). Overall, these results show that by directly optimizing for semantic diversity through goal-directed exploration, ACES variants generate problems that are more semantically diverse, but also generally more diverse.

**More challenging** Figure 3d shows the evolution of the average fitness of the puzzles generated over the last 5 iterations (800 attempted puzzle generations). Here we see the ability of ACES variants to continuously generate increasingly challenging puzzles across the experiment, while ELM variants saturate earlier on. StaticGen, a static generation method, consistently generates puzzles with low fitness. Figure 3f shows the distribution of fitness of the whole archive at the end of experiments. These distributions are all bimodal, with problems having either very low fitness (solved most of the time) or very high fitness (solved only a couple times out of the 50 solving attempts). While StaticGen mostly generates simple problems (peak in the lowest fitness bin), all the algorithm optimizing for difficulty (all others) seem to do so efficiently — showing peaks in the highest fitness bin.

Finally, Figure 3e shows the evolution of *QD scores*, a metric used to evaluate QD methods that sums the fitness of the very best solution found in each filled niche. This metric captures both diversity (more niches means more terms in the sum) and local quality (higher quality solutions means higher terms in the sum). Here again, ACES variants significantly outperform other baselines. All these results are evidence that ACES variants efficiently produce a larger diversity of more challenging problems than existing algorithms.

**Finetuning** We conducted finetuning experiments using the Llama-3-8b model with datasets generated by WizardCoder (a variant of the state-of-the-art method WizardLM for generating synthetic data[Luo et al., 2023, Xu et al., 2023]), StaticGen (established baselines), and our proposed ACES-ELM method. We then evaluated the model's performance using the greedy pass@1 metric on a series of test sets. These test sets were equally composed of puzzles from our method and StaticGen with increasing difficulty levels, generated using a seed different from the training data.

Results revealed that the Llama-3-8b model finetuned with ACES-ELM-generated data achieved a remarkable pass@1 score of 53.3 on the most challenging test set (see Figure 4b), significantly outperforming both baseline methods and the Llama-3-70B model. While models finetuned with WizardCoder and StaticGen showed improvements over the baseline (achieving pass@1 scores of 49.4 and 41.6, respectively), they consistently underperformed compared to the ACES-ELM-trained model. Notably, the performance gap between ACES-ELM and baseline methods widened as testset difficulty increased, demonstrating ACES-ELM's effectiveness in preparing models for complex coding challenges and generating high-quality training data.

## 4.4 ACES scales with LLM size

**Diversity** We found that ACES could generate more diverse and challenging problems, but what happened if we tried to use a larger LLM? We compare ACES archive generated using Llama-3-70b against larger models (Llama 405B model and Mistral large 2). Using those LLMs leads to a better Quality-Diversity score overall (see Figure 5), up to 25.6% for Mistral Large and 12.3% for Llama 405B (using Llama 405B and Mistral large 2 both for the difficulty metric and skill labeling). This demonstrates how ACES scales with models of larger size.

**difficulty** Evaluating Mistral Large and Llama 405B on each other's generated datasets reveals pass@1 scores of 56.7% and 58%, respectively (Figure 4a). These results demonstrate the effectiveness of our method in creating challenging benchmarks, even for state-of-the-art models, and highlight the transferability of difficulty measures across models with similar capabilities.

Furthermore, when tested on the archive generated by Llama-3-70B using ACES-ELM, Mistral Large 2 achieved a 70% pass@k score, while Llama-3-405B reached 74%. In contrast, Llama-3-70B itself attained a pass@1 score of 36.8%. These findings underscore the superior problem-solving

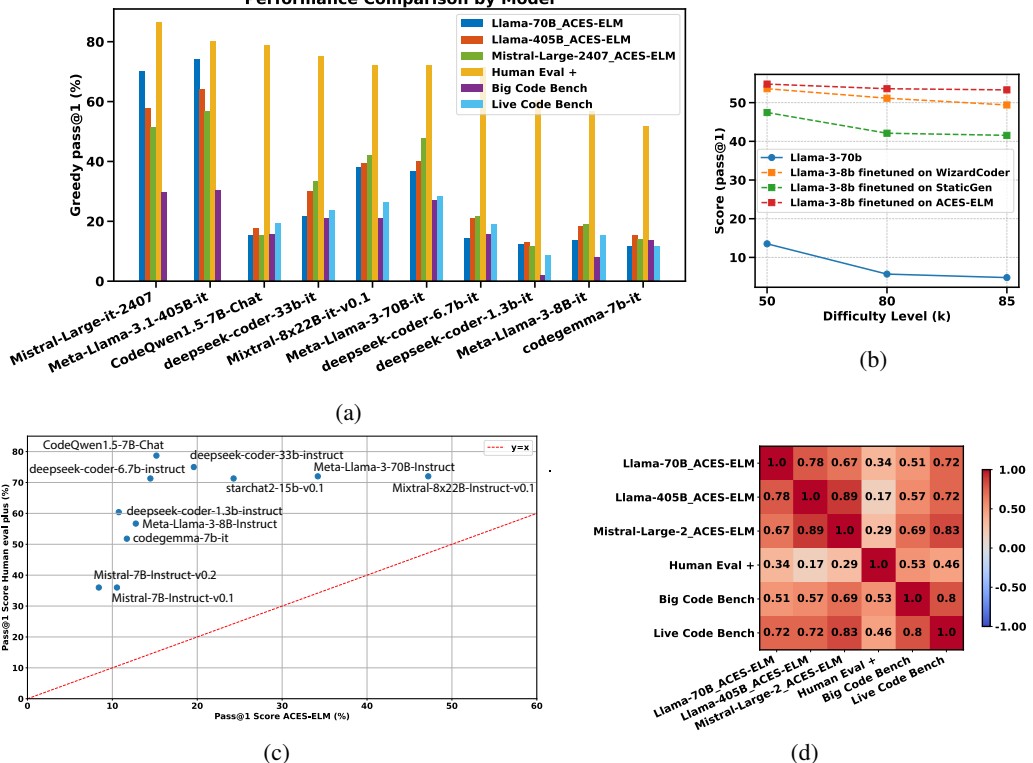

(a)

(b)

(c)

(d)

Figure 4: **(a)** Greedy pass@1 scores for various models on datasets generated by ACES-ELM with Llama-3-70b, Llama-3-405B, Mistral, on the HumanEval + benchmark, and on recent benchmark created to limit contamination and saturation compared to HumanEval (BigCodeBench and Live-CodeBench). Scores not found are represented as 0. **(b)** Greedy pass@1, after finetuning Llama-8b on archive generated by WizardCoder, StaticGen, and ACES-ELM, on a series of test sets equally composed of mixtures of puzzles from ACES-ELM and StaticGen. See rebuttal main text for additional details. **(c)** Pass@1 competence of all models on *HumanEval+* versus on problems generated by our best problem generator (ACES-ELM). Problems that are more challenging for a model are more generally challenging for others too (similar ranks in (a)). Models in the top left corner of (b) may be overfitting to HumanEval+, as their strong performance there do not translate into higher performance on our generated problems. **(d)** Correlation matrix of pass@1 scores of the different datasets. Our method achieves a correlation up to 0.83 with Live Code Bench and 0.69 with Big Code Bench, whereas HumanEval has only a correlation of 0.46 and 0.53 respectively.

capabilities of Mistral Large and Llama-405B. However, it's noteworthy that even these advanced models have not saturated the benchmark generated by Llama-3-70B, as there remains approximately 30% room for improvement to solve the benchmark fully.

## 4.5 ACES generates more challenging problems than the ones found in existing benchmarks

ACES variants generate problems that are more challenging than problems generated by other baselines. But how challenging are they really? Here, we measure the competence of 10 state-of-the-art LLM problem solvers and compare it to the performance of these same models on existing human-curated benchmarks. Figure 4a reports the pass@1 scores for 10 LLM solvers over two existing programming puzzles benchmarks: HumanEval + [Liu et al., 2024], Big Code Bench, Live Code Bench as well as over the set of problems generated by ACES variants (with Llama-3-70b, Llama-3.1-405b and Mistral Large 2) and other baselines in the experiments presented above. Results for additional models can be found in Appendix Figure 8. Overall, our generated sets are more challenging across models than the HumanEval sets usually used to benchmark code LLMs. *CodeQwen1.5-7B-Chat*, *Llama3-70B-instruct* and *Mixtral-8x22B-Instruct-v0.1*, one of the best open source models on HumanEval with 78.7, 72 and 72% pass@1 respectively, get a considerable drop in pass@1, falling to 15, 47 and 34% respectively on problems generated by ACES-ELM with Llama-3-70b (Figure 4c).

Our findings demonstrate the transferability of puzzle difficulty across various language models. Models exhibiting lower scores on our datasets generated with Llama 70B also have lower scores on the datasets generated by Llama 405B and Mistral, as well as on LiveCodeBench [Jain et al., 2024] (which is specifically designed to avoid contamination) and BigCodeBench [Zhuo et al., 2024]. Figure 4a illustrates this correlation, with HumanEval being the notable exception. This anomaly suggests potential contamination in models optimized for the HumanEval benchmark, particularly in smaller models. Figure 4d presents the correlation between pass@1 scores across all datasets, averaged over models. Our method has generated benchmarks that correlate more with uncontaminated benchmarks (LiveCodeBench and BigCodeBench) than HumanEval. Specifically, the Mistral-generated dataset using ACES-ELM achieved a correlation of 0.83 with LiveCodeBench. In contrast, HumanEval only reached a correlation 0.46 with Live Code Bench, underscoring the challenging nature and value of our generated benchmark for comparing LLMs.

## 5  Discussion

This paper presented ACES, an autotelic generative algorithm designed to generate a diversity of challenging Python programming puzzles. Our experiments showed that ACES generates problems that are both more diverse and more challenging than the ones generated by existing generative approaches or the ones found in human-curated benchmarks.

**Limitations and improvements**   ACES has several limitations. The algorithm relies on several assumptions: 1) the labeling should be accurate, 2) the generator should be good at reaching its goals, 3) the correlation between the feature (descriptors and quality) of parent and children problems should be sufficiently high (heredity). Although these assumptions are only partially verified (see Section 4.2 and Appendix Figures 9 and 10), our results show that these are *good enough* for allowing the effective optimization of diversity and difficulty in our domain. We expect that progress along the three lines can be made by harnessing the increasing capabilities of future language models and will automatically translate into higher performing autotelic generative models of problems. The heredity property presents an interesting research question: in principle, it underlies all evolutionary methods making use of LLMs but, in practice, it has not been evaluated or discussed [Bradley et al., 2023a,b].

**Creating and releasing challenging LLM benchmarks**   ACES allows for the automatic generation of programming puzzles tailored to LLM's current capabilities and we demonstrated that we generate puzzle sets that are more challenging than current code evaluations. This brings hope for generating tomorrow's next generation of code benchmarks, as today's evaluations are almost saturated. However, important steps need to be performed before this can be done. First, as is done in Schuster et al. [2021], we currently only test solutions with one set of arguments, there should be tests with a wide range to measure solution robustness. Empirical difficulty is not the only thing to be expected of exercises: the puzzles need to make sense and be hard for the good reasons. This requires a more complete picture of a puzzle's quality than difficulty alone, which remains an open problem. Overall, we believe that open-ended algorithms will play an increasing role in automatically evaluating LLMs in the future [Samvelyan et al., 2024].

**Other applications**   ACES is a general algorithm that can be easily translated to other application domains by letting the user define a new descriptor and quality function. Swapping the difficulty metric with a more subjective objective estimated by an LLM could let ACES generate problems adapted for human students of a specific level in educational contexts, for example. In artificial intelligence, one could envision a self-play loop where a learning agent iteratively generates a diversity of problems maximizing its current learning progress (quality metric), then trains on them to augment its competence [Sukhbaatar et al., 2017, Silver et al., 2017, Colas et al., 2022].

**Broader impacts**   Open-ended exploration algorithms in general have wide-ranging implications when scaled up. They could potentially help in discovering harmful artifacts (in code domains, harmful bots or cyberattack programs), as well as help find solutions or red-team existing systems. We note that an evolutionary optimization algorithm such as ACES needs extensive fitness feedback, either limiting its use by bad actors or making them detectable. Positive applications of open-ended exploration algorithms applied to problem generation range from educational technologies to automated scientific discovery, helping shoulder some of the rising needs of the future.

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

# A    Appendix

## A.1    Additional Related Work

**Descriptor spaces for exploration methods**    In all open-ended exploration methods, one must define a Behavioral Characterization (BC) space to characterize novelty. The earliest works used predefined low-dimensional descriptors to represent generated artefacts [Lehman and Stanley, 2011a, Baranes and Oudeyer, 2013, Mouret and Clune, 2015], which constrains the search along a handful of features one can code a descriptor for. More recent works have relied on higher-dimensional learned or pretrained embedding functions [Nair et al., 2018, Laversanne-Finot et al., 2018, Reinke et al., 2020], and even hierarchies of such spaces, each representing different perceptual features of the generated artefacts [Cully and Demiris, 2018b, Etcheverry et al., 2020]. Diversity-search algorithms sometimes need to be adapted to work with such high-dimensional spaces whose discretization leads to an exponential number of cells [Vassiliades et al., 2017]. But the main issue is that they are hardly interpretable and might not always align with the dimensions of variation humans find meaningful. With ACES, we propose an autotelic diversity-producing algorithm that constrains the search along a set of abstract, interpretable and hard-to-compute features of interest evaluated by LLMs.

## A.2    Prompts

Here are the various prompts we use for ACES and all baselines.

**Skills description.**    Skills description used to label problem. see prompt A.2

---

**List of semantic descriptors**

0 - **String Manipulation**
1 - **Mathematical Operations**
2 - **Conditional Logic**
3 - **Recursion**
4 - **Brute Force Search**
5 - **Dynamic Programming**
6 - **Greedy Algorithms**
7 - **Backtracking**
8 - **Set Operations**
9 - **Permutations and Combinations**
10 - **Probability and Statistics**
11 - **Pattern Recognition**
12 - **Sorting and Ordering**
13 - **Binary Operations (bitwise shifting, AND, OR)**
14 - **Geometry and Coordinate Manipulation**
15 - **Algorithm Optimization**
16 - **Number Theory (factors, primes, etc.)**
17 - **Graph Theory (paths, edges, vertices)**
18 - **Array Indexing**
19 - **Hashing**

---

**Example of puzzle labelling**

**Puzzle to label:**

```
def f(s: str) -> bool:\\
    """Find a string with 1000 'o's but no two adjacent 'o's.
        """\\
    return s.count('o') == 1000 and s.count('oo') == 0\\
def g():\\
    return ('h' + 'o') * 1000\\
assert f(g())\\
```

**Assistant:** To solve the problem, we need to understand the logic of the 'f' function and how it checks for the presence of 1000 'o's and no adjacent 'o's. We also need to understand the implementation of the 'g' function, which generates the string with 1000 'o's.

Based on this understanding, the necessary programming skills are:
- 1: Counting and combinatorics (to understand the counting of 'o's and 'oo's in the string)
- 5: String Manipulation (to understand the string concatenation and searching)

Therefore, the list of indices for the problem is: [1, 5]

**Prompt for the puzzle generator of ACES.** This prompt is used for ACES and all autotelic variants.

---

**ACES**

**User:**
Consider Python Programming Puzzles (P3). P3 consists of two functions: a problem function 'f' and its corresponding solution 'g'. The challenge lies in constructing a SAT problem 'f' and a function 'g' such that 'f(g())' evaluates to 'True'

**Main Rules:**
- Each puzzle includes two functions: 'def f(...)' and 'def g(...)'.
- The first argument of 'f' is always the output from 'g()'.
- Ensure 'f' and 'g' have matching argument signatures (e.g., 'def f(solution, arg1=value1, arg2=value2, ...)' and 'def g(arg1=value1, arg2=value2, ...)'). You also need to set the value of argument of f (arg1,arg2,...) and g when you define them.
- Avoid using 'f' inside 'g', and 'g' inside 'f'.
- Include any necessary imports so your code runs smoothly.
- Give a clear Puzzle description that must be brief and diverse compared to the other puzzles.
- Make sure the puzzle is self-contained within these two functions.
- Make sure that each puzzle have just all required skills (see below)

**P3 Format:**
Puzzle description: A two to four sentence summary of the puzzle's content. To explain what is the problem 'f', and how you can solve it with 'g'.

```python
def f(solution, args=...) -> bool:
    # Python code to test the solution returned by g.
    # This function is a test unit and must return True if the
        solution is correct, and False otherwise.

def g(args=...) -> solution:
    # Python code to generate a solution for the problem.
    # The solution should generalize to all possible args.
    return solution

assert f(g()) == True
```

**Examples:**

Puzzle 0:
Puzzle description: [puzzle description]

- Difficulty score: [puzzle score] out of 100

- This puzzle has the following skills:
[skillslist]

[Python Programming Puzzle]

Puzzle 1:
...
Puzzle 2:
...

Generate 5 P3 similar to previous Examples. Ensure that all new puzzles are more challenging than Puzzle from previous examples.
You should aim to generate puzzles with a Difficulty score between 90 and 100 out of 100.

**Please make sure that new puzzles have JUST ALL the following skills**:
[list target skills]

**New 5 problems:**
**Assistant:**

---

**ACES-ELM**

**User:**
Consider Python Programming Puzzles (P3). P3 consists of two functions: a problem function 'f' and its corresponding solution 'g'. The challenge lies in constructing a SAT problem 'f' and a function 'g' such that 'f(g())' evaluates to 'True'

**Main Rules:**
- Each puzzle includes two functions: 'def f(...)' and 'def g(...)'.
- The first argument of 'f' is always the output from 'g()'.
- Ensure 'f' and 'g' have matching argument signatures (e.g., 'def f(solution, arg1=value1, arg2=value2, ...)' and 'def g(arg1=value1, arg2=value2, ...)'). You also need to set the value of argument of f (arg1,arg2,...) and g when you define them.
- Avoid using 'f' inside 'g', and 'g' inside 'f'.
- Include any necessary imports so your code runs smoothly.
- Give a clear Puzzle description that must be brief and diverse compared to the other puzzles.
- Make sure the puzzle is self-contained within these two functions.

- Make sure that each puzzle have just all required skills (see below)

**P3 Format:**
Puzzle description: A two to four sentence summary of the puzzle's content. To explain what is the problem 'f', and how you can solve it with 'g'.

```python
def f(solution, args=...) -> bool:
    # Python code to test the solution returned by g.
    # This function is a test unit and must return True if the
        solution is correct, and False otherwise.

def g(args=...) -> solution:
    # Python code to generate a solution for the problem.
    # The solution should generalize to all possible args.
    return solution

assert f(g()) == True
```

**Examples:**

Puzzle 0:
Puzzle description: [puzzle description]

- Difficulty score: [puzzle score] out of 100

- This puzzle has the following skills:
[skills list]

[Python Programming Puzzle]

Puzzle 1:
...
Puzzle 2:
...

Generate 5 P3 similar to the last Examples (Puzzle 2). Ensure that all new puzzles are more challenging than Puzzle 2.
You should aim to generate puzzles with a Difficulty score between 90 and 100 out of 100.

**Please make sure that new puzzles have JUST ALL the following skills**:
[list target skills]

**New 5 problems inspired by Puzzle 2:**
**Assistant:**

**Prompt for the puzzle generator of Static gen.**

**Static gen**

**User:** Consider Python Programming Puzzles (P3). P3 consists of two functions: a problem function 'f' and its corresponding solution 'g'. The challenge lies in constructing a SAT problem 'f' and a function 'g' such that 'f(g())' evaluates to 'True'

**Main Rules:**
- Each puzzle includes two functions: 'def f(...)' and 'def g(...)'.
- The first argument of 'f' is always the output from 'g()'.

- Ensure 'f' and 'g' have matching argument signatures (e.g., 'def f(solution, arg1=value1, arg2=value2, ...)' and 'def g(arg1=value1, arg2=value2, ...)'). You also need to set the value of argument of f (arg1,arg2,...) and g when you define them.
- Avoid using 'f' inside 'g', and 'g' inside 'f'.
- Include any necessary imports so your code runs smoothly.
- Give a clear Puzzle description that must be brief and diverse compared to the other puzzles.
- Make sure the puzzle is self-contained within these two functions.

**P3 Format:**
Puzzle description: A two to four sentence summary of the puzzle's content. To explain what is the problem 'f', and how you can solve it with 'g'.

```python
def f(solution, args=...) -> bool:
    # Python code to test the solution returned by g.
    # This function is a test unit and must return True if the
        solution is correct, and False otherwise.

def g(args=...) -> solution:
    # Python code to generate a solution for the problem.
    # The solution should generalize to all possible args.
    return solution

assert f(g()) == True
```

**Examples:**

Puzzle 0:
Puzzle description: [puzzle description]

- Difficulty score: [puzzle score] out of 100

- This puzzle has the following skills:
[skillslist]

[Python Programming Puzzle]

Puzzle 1:
...
Puzzle 2:
...

Generate 5 different P3 similar to previous Examples.

**New 5 problems:**
[list target skills]

**New 5 problems inspired by Puzzle 2**
**Assistant:**

**Prompt for the puzzle generator of ELM and ELM semantic.** This prompt is used for non-autotelic baselines.

**User:** Consider Python Programming Puzzles (P3). P3 consists of two functions: a problem function 'f' and its corresponding solution 'g'. The challenge lies in constructing a SAT problem 'f' and a function 'g' such that 'f(g())' evaluates to 'True'

**Main Rules:**
- Each puzzle includes two functions: 'def f(...)' and 'def g(...)'.
- The first argument of 'f' is always the output from 'g()'.
- Ensure 'f' and 'g' have matching argument signatures (e.g., 'def f(solution, arg1=value1, arg2=value2, ...)' and 'def g(arg1=value1, arg2=value2, ...)'). You also need to set the value of argument of f (arg1,arg2,...) and g when you define them.
- Avoid using 'f' inside 'g', and g inside 'f'.
- Include any necessary imports so your code runs smoothly.
- Give a clear Puzzle description that must be brief and diverse compared to the other puzzles.
- Make sure the puzzle is self-contained within these two functions.

**P3 Format:**
Puzzle description: A two to four sentence summary of the puzzle's content. To explain what is the problem 'f', and how you can solve it with 'g'.

```python
def f(solution, args=...) -> bool:
    # Python code to test the solution returned by g.
    # This function is a test unit and must return True if the
        solution is correct, and False otherwise.

def g(args=...) -> solution:
    # Python code to generate a solution for the problem.
    # The solution should generalize to all possible args.
    return solution

assert f(g()) == True
```

**Examples:**

Puzzle 0:
Puzzle description: [puzzle description]

- Difficulty score: [puzzle score] out of 100

- This puzzle has the following skills:
[skillslist]

[Python Programming Puzzle]

Puzzle 1:
...
Puzzle 2 (to mutate):
...

Generate 5 P3 similar to the last Examples (Puzzle 2). Ensure that all new puzzles are more challenging than Puzzle 2.
You should aim to generate puzzles with a Difficulty score between 90 and 100 out of 100.

**Please make sure that new puzzles have JUST ALL the following skills**:
[list target skills]

**New 5 problems inspired by Puzzle 2**
**Assistant:**

---

### Description prompt

**User:** A Python programming puzzle is defined by two functions, the problem f(solution, arg1=value1, arg2=value2, ..) and the solution. f defines an algorithmic puzzle, and the solution solves this puzzle.

You should pay a particular attention that the puzzle is solved if and only if **f(solution) == True**.

Your role is to write a one or two sentence the description of the puzzle's goal (what the solution should be), remember that the solution that satisfy the goal must be given as the first argument of 'f'.

You can start by: 'Find the solution: "arg solution" (describe its type shortly) that should (here you should speak about the solution: "arg solution" and how it should solve all the constraints of the puzzle with respect to others args (describe their types shortly)) ...'.

For example:

'Given a string 'str1', find the length of the longest substring without repeating characters.'

'Given two sorted arrays 'nums1' and 'nums2' of size 'm' and 'n' respectively, return the median of the two sorted arrays.'

The puzzle is:

...

**Assistant:**

---

### Skill labeling prompt

**User:** You are a helpful assistant to a Professor teaching a programming course in Python.
The Professor want to give Pyhton programming puzzles to his Computer Science student to teach them Python.

A Python programming puzzle is defined by two functions, the puzzle f(...) and the solution g(...). f defines an algorithmic challenge, and g solves this challenge. g is a solution to f if and only if f(g()) == True.

The Professor want to evaluate the diversity of those puzzles, can you label the following puzzle given the following list of topics, please?

The list of topics is:

skills list

The puzzle is:

...

Respond with two or three sentence explaining the topics used in the puzzle.

Then summarize your response by giving a list from 1 to 5 index corresponding to topics that are actually used in the puzzle above in this format: 'The list of skill use is: [].' where [] is the list of index of the topics used in the puzzle for example [3,5,6].

**Assistant:**

---

### Solver prompt

**User:** You will be given a function. Respond only in code with a correct, efficient implementation of the function. You will need to generate the correct solutions (g), for the Problem 2 that satisfies the condition f(g()) == True.

Problem 0:
def f(...

---

Solution 0:
def g(...
Problem 1:
def f(...
Solution 1:
def g(...

Now you need to give the solution (def g(...)) to the following Problem 2 that satisfies the condition $f(g()) == True$.

Problem 2:
def f(...

**Assistant:**

---

**Puzzle to mutate:**

```python
from typing import*
def f(n: int, lst=['apple', 'banana', 'orange', 'grape']) ->
    bool:
    """Check if the given element n is a prefix of any element
        in the list lst"""
    for word in lst:
        if word.startswith(n):
            return True
    return False

def g(lst=['apple', 'banana', 'orange', 'grape']):
    return lst[1]

assert f(g())
```

**Assistant:**

```python
from typing import*
import math
def f(n: int, lst=[1, 2, 3, 4, 5]) -> bool:
    """Check if the given list contains any perfect square
        number and if it is divisible by n."""
    for num in lst:
        if math.isqrt(num)**2 == num:
            return n % num == 0
    return False
def g(lst=[1, 2, 3, 4, 5]):
    for num in lst:
        if math.isqrt(num)**2 == num:
            return num
    return None
assert f(g())
```

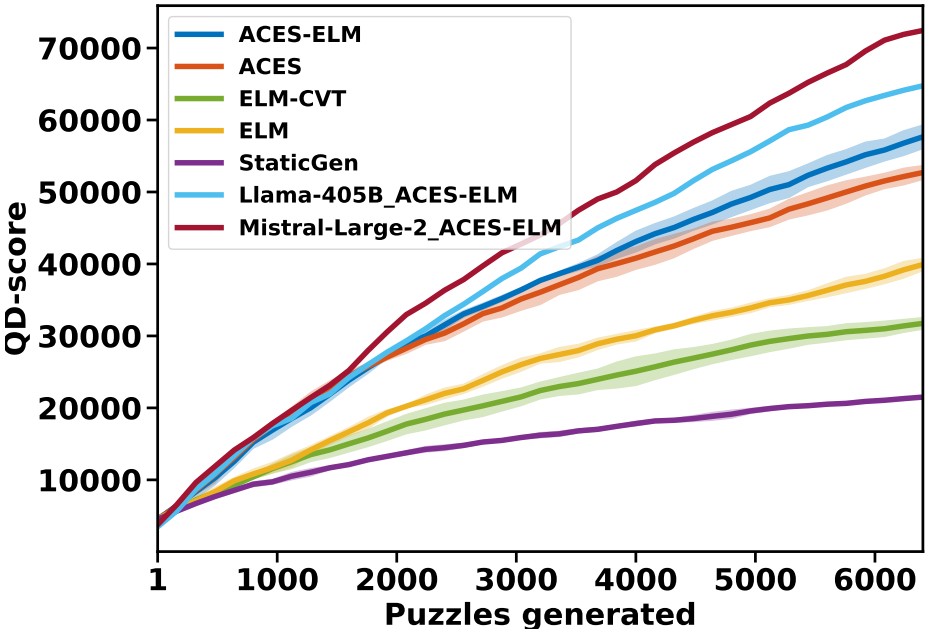

(a)

Figure 5: **(b)** Quality-Diversity (QD) score updated figure with ACES-ELM using state-of-the-art open weight LLM (Llama-3-405B and Mistral Large 2)

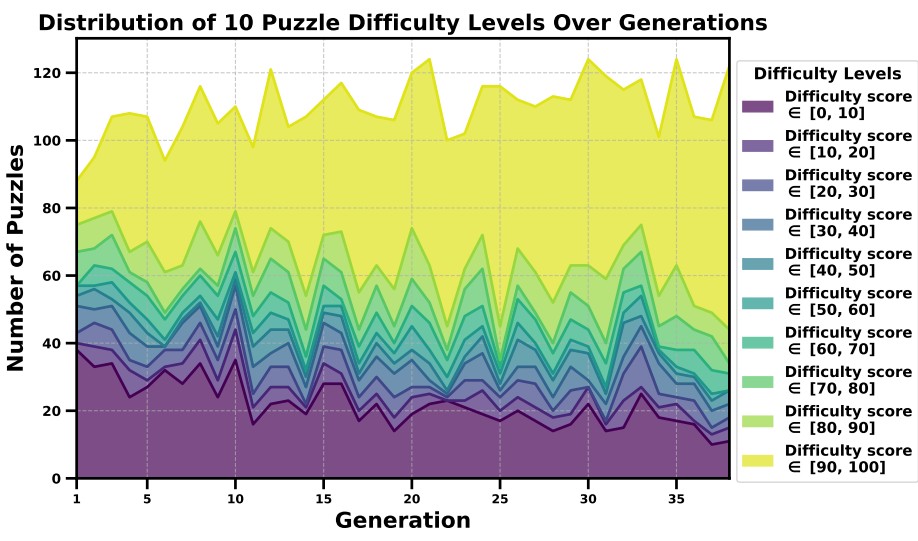

(a)

Figure 6: **(e)** Evolution of Puzzle Difficulty Distribution generated by ACES: Decile Analysis of Generated Puzzles over generations.

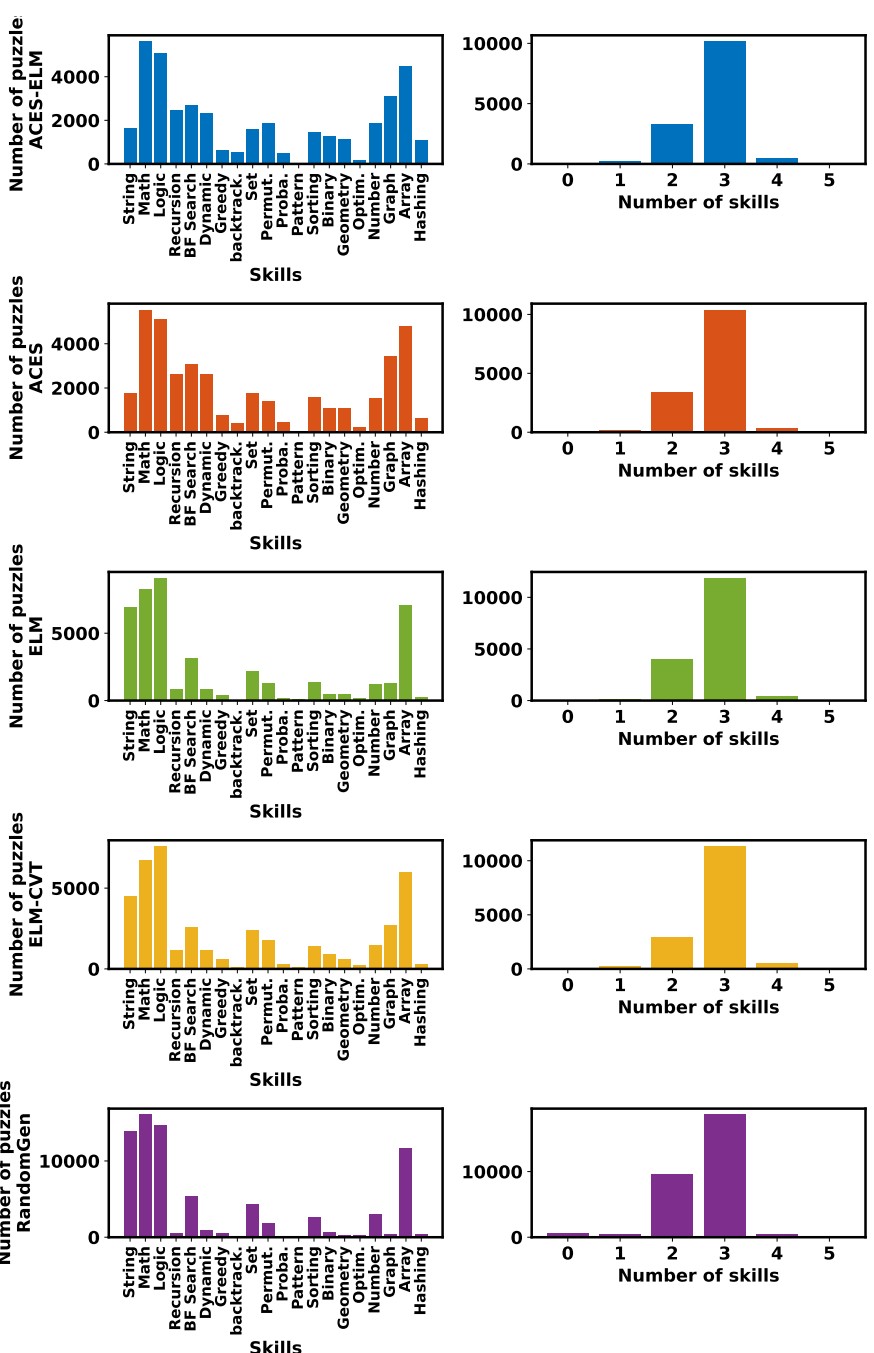

Figure 7: Distribution of labeled skills (left) and number of skills labeled (right) for all algorithms (rows).

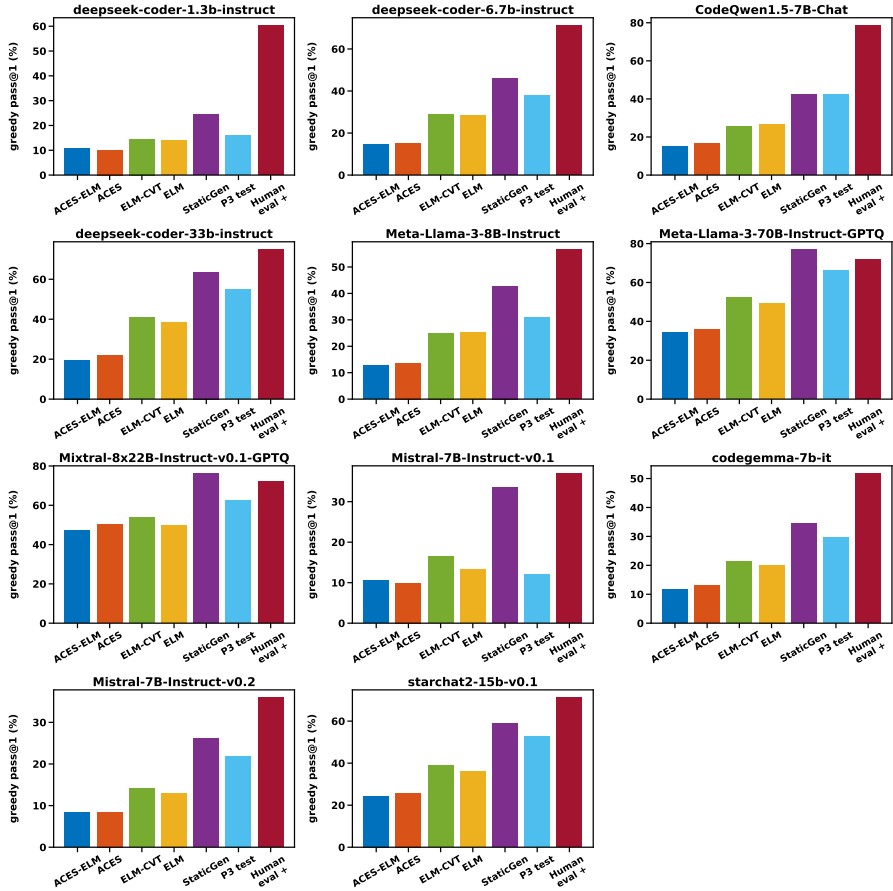

Figure 8: Pass@1 competence of various state-of-the-art problem solvers on existing benchmarks (*HumanEval+* and *P3 test*) and on problems generated by our algorithms.

### A.3 Semantic descriptor diversity

### A.4 Performance Comparison

### A.5 Heridity

### A.6 Examples of generated puzzles

In this section, we present a few puzzles and solutions generated by our different methods and examine them qualitatively.

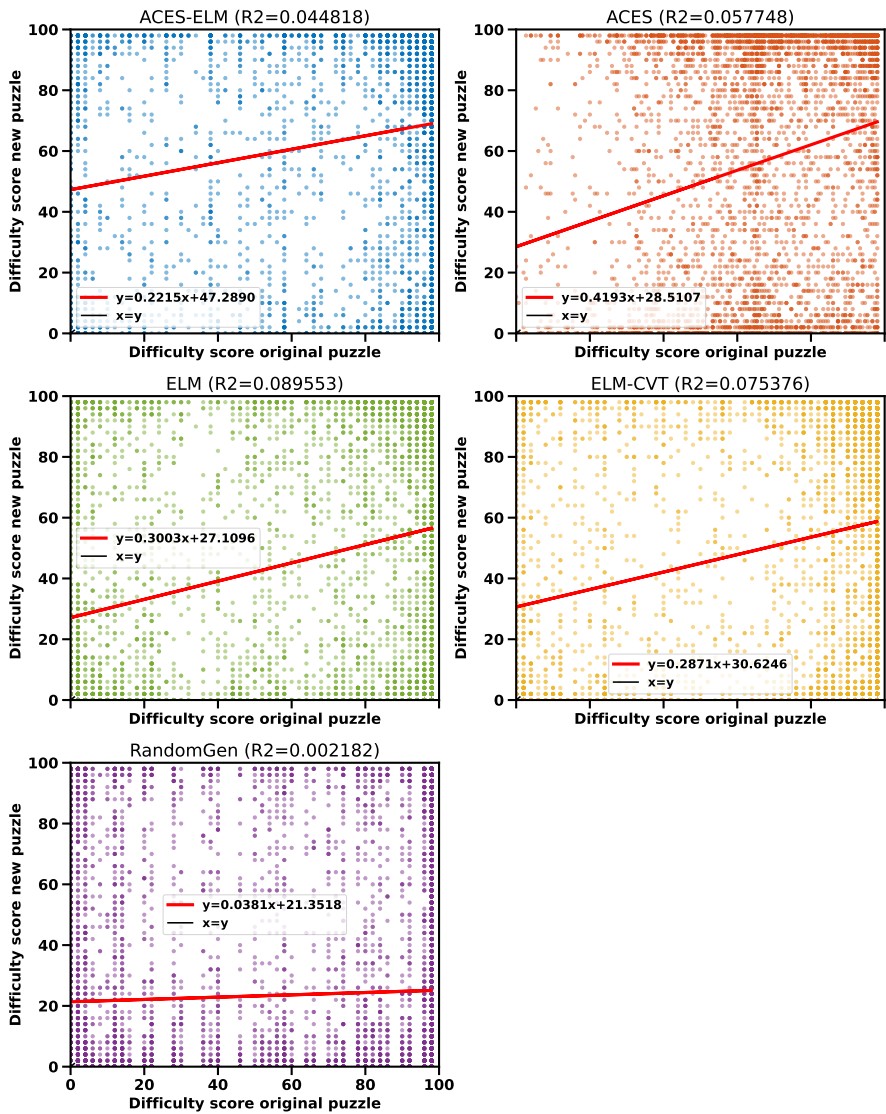

Figure 9: Evolution of the difficulty score from puzzles used as a few-shot example to generated puzzles based on those previous puzzles.

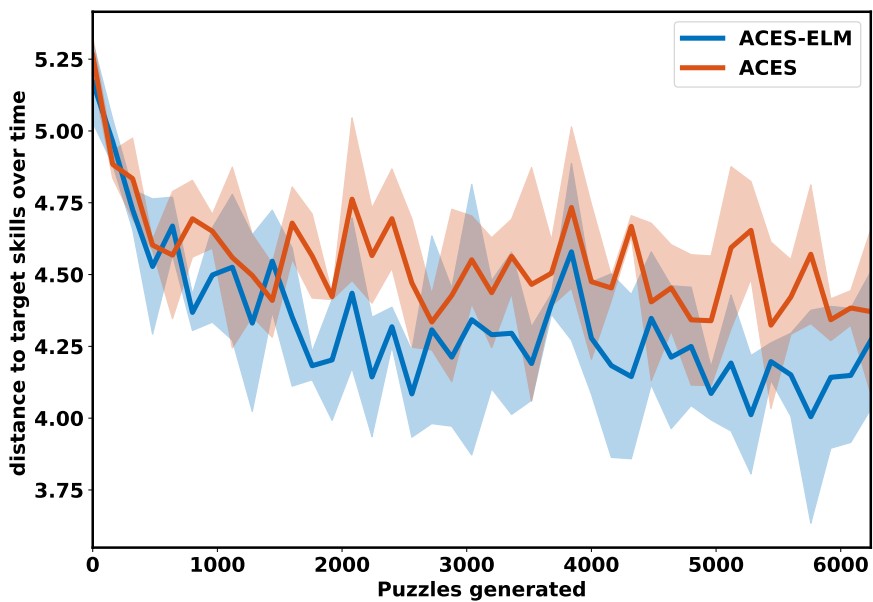

Figure 10: Distance to target skills over time.

**Example generation for ACES**

```python
from typing import List
def f(moves: List[List[int]], initial_state=[3, 3, 2, 2, 3,
    8]) -> bool:

    def bot_move() -> bool:
        vals = sorted(state, reverse=True)
        i_largest = state.index(vals[0])
        state[i_largest] -= max(vals[0] - vals[1], 1)
    state = initial_state[:]
    for (i, n) in moves:
        assert 0 < n <= state[i], 'Illegal move'
        state[i] -= n
        if set(state) == {0}:
            return True
        assert any(state), 'You lost!'
        bot_move()
def g(initial_state = [3, 3, 2, 2, 3, 8]):

    state = initial_state[:]
    moves = []

    def bot_move():  # bot takes objects from the largest heap
        to make it match the second largest heap
        vals = sorted(state, reverse=True)
        i_largest = state.index(vals[0])  # largest heap
        state[i_largest] -= max(vals[0] - vals[1], 1)  # must
            take some, take 1 in case of tie

    def losing(h):  # return True if h is a losing state
        xor = 0
        for i in h:
            xor ^= i
        return xor == 0

    def optimal_move():
        assert not losing(state)
        for i in range(len(state)):
            for n in range(1, state[i] + 1):
                state[i] -= n
                if losing(state):
                    moves.append([i, n])
                    return
                state[i] += n
        assert False, "Shouldn't reach hear"

    while True:
        optimal_move()
        if max(state) == 0:
            return moves
        bot_move()
```

