# OpenReview forum: "ACES: Generating a Diversity of Challenging Programming Puzzles with Autotelic Generative Models"
_NeurIPS.cc/2024/Conference — NeurIPS 2024 spotlight_

### Official Review · Reviewer_ESVE · 2024-07-05

**Soundness:** 4
**Presentation:** 3
**Contribution:** 2
**Rating:** 6
**Confidence:** 3

**Summary:**

The paper presents Autotelic CodE Search (ACES), a method for generating diverse and challenging Python programming puzzles using state-of-the-art generative models. Inspired by Quality-Diversity, ACES aims to automate the problem generation process by optimizing for both diversity and difficulty. ACES iteratively prompts an LLM to generate puzzles by using previously generated problems as in-context examples. The generated puzzles are more diverse and challenging than those produced by baseline methods and existing benchmarks, as evaluated across 11 state-of-the-art code LLMs.

**Strengths:**

- The presentation, writing and clarity of the paper are great.

- The method creatively combines evolutionary algorithms (especially Quality-Diversity algorithms) with LLMs to generate programming puzzles that are both diverse and challenging.

- The proposed approach is simple and effective at generating a diversity of challenging puzzles. Experiments showed that ACES generates problems that are both more diverse and more challenging than the ones generated by existing generative approaches or the ones found in human-curated benchmarks.

**Weaknesses:**

- My main concern is about the contribution. What are the practical applications of this framework? Why is it valuable to generate a diversity of challenging puzzles? Demonstrating some downstream tasks would be helpful to justify the importance of generating diverse and challenging puzzles. For example, showing how fine-tuning a model on the newly generated data can improve performance would provide a clear motivation.

- The paper states, "Puzzles for which no solution is found are considered invalid and are discarded." I believe this is a limitation because the primary goal of generating new data is likely to enhance training. By only considering puzzles that the target LLM solver can solve, we limit the difficulty of the generated puzzles. This, in turn, restricts the potential benefits of using this new data for training and improving models.

**Questions:**

- What are the practical applications of this framework? Why is generating diverse and challenging puzzles an important problem to address?

- I am uncertain if the difficulty measure is appropriate. I would expect the difficulty measure to be very close to 0 for problems that the target LLM solver cannot solve and very close to 100 for problems it can solve, with almost nothing in-between. Can you provide more details on this aspect?

- What does the shaded area represent on the graphs?

- I don't understand Figure 3(f). The x-axis represents fitness rescaled between 0-100, but what does the color signify?

- OMNI-EPIC [1] utilizes a task archive in a manner similar to your puzzle archive. I believe you should cite this paper to acknowledge the related work appropriately.

[1] OMNI-EPIC: Open-endedness via Models of human Notions of Interestingness with Environments Programmed in Code

# Minor Corrections

L167: There is an inappropriate exclamation mark in "Appendix Section! A.2."

**Limitations:**

Yes, the authors adequately addressed the limitations.

---

> ### Author Rebuttal · Authors · 2024-08-06
>
> We thank Reviewer ESVE for their review, and especially for appreciating the clarity of the paper, and the simplicity and effectiveness of the method. We hope to be able to convince them of the promise of the method for practical applications.
>
> > What are the practical applications of this framework?
>
> * Generality of the framework. We understand that the formulation of our aims in this paper might seem a bit general (creating sets of diverse and difficult problems) but the applications are quite promising in our opinion! First, ACES transfers to any domain where solutions to problems can be exactly evaluated and where LLMs already have knowledge of several categories of problems (as also noted by Reviewer qqHU). This includes important tasks like formal math or RL environments that can described with code.
>
> * Creating benchmarks. Within the domain we have chosen (programming puzzles), the target task we have in mind for ACES is creating LLM evaluation sets. There are other considerations when creating a benchmark than difficulty and diversity, but getting these right is a necessary condition for it to be useful. Distilling our generated sets into trusted benchmarks usable by the community would require some additional curation, but we believe it to be well within reach.
>
> * Creating finetuning data. In our additional results (Figure 1(d)) we present results of finetuning models with our generated data (see also general response). On tests sets created with a mixture of StaticGen and ACES-ELM data, finetuning on (distinct) ACES-ELM generated data yields strong and consistent improvements compared to finetuning on (distinct) StaticGen-generated data. This is good evidence of increasing capabilities when finetuned on our data.
>
> * Generating open-ended tasks is an important goal in itself. This is precisely the aims of open-ended exploration algorithms (such as OMNI-EPIC you mention later). ACES belongs to that line of works, applies it to an important domain, and contributes algorithmic improvements. Reviewer LkwK also notes that this is an important fundamental topic in itself, beyond the practical applications.
>
> > [...] I believe this is a limitation because the primary goal of generating new data is likely to enhance training.
>
> We agree! This is a limitation of our current difficulty measure and have added additional discussion in the Limitations section of the paper. This kind of difficulty metric is used to train open-ended agents in other settings. For instance, in Goal-GAN [1], a goal-conditioned RL agent targets position goals that are difficult but solvable, since unreachable positions usually give no feedback at all. This approach is widespread in intrinsically-motivated RL [2, 3]. In the classic case of self-play, the opponent provides feedback by being neither too easy nor too hard to beat by construction. For ACES to implement similar dynamics (and lead to the discovery of puzzles that are truly unsolvable from the point of view of the solver we start with yet solvable in general) one should finetune the solver as the data gets generated. This would not require modifying the difficulty metric we use. We have added additional discussion in the Discussion section.
>
> ### Questions
>
> > I don't understand Figure 3(f).
>
> You are right that this figure could be clearer. This is a figure representing the distributions of difficulties, in discrete fitness bins, for each method, at the end of the generation process. The color represents the number of puzzles whose difficulty falls into each bin (color histogram). The takeaways from this figure are that these distributions are quite peaked towards 0 and 100 and that ACES and ACES-ELM have bigger peaks towards 100 than StaticGen and ELM.
>
> > I am uncertain if the difficulty measure is appropriate. [...] Can you provide more details on this aspect?
>
> We first recall that high difficulty/fitness (close to 100) corresponds to hard problems (solved once out of 50), and difficulty close to 0 corresponds to easy problems (solved 50 times out of 50).
> The motivation for this metric comes from observations around pass@k. Pass@k was introduced in the Codex paper [4] and measures the proportion of puzzles which have at least 1 solution found in k solver attempts. It has become a standard measure for evaluating code (or math) models. The interesting part is that as you increase k (with nonzero temperature), pass@k continues increasing logarithmically with k (see [5], Figure 7 for instance). Repeatedly sampling from your LLM is usually an effective, but costly, way of getting solutions to hard problems [6, 7]. To recap: puzzles do not neatly fall in solvable/unsolvable categories; some of them are solved only once every 50 attempts on average. These are the kinds of puzzles we want to generate.
>
> In our experiments, as outlined in our previous comment, the difficulty distribution is close to 0 for the StaticGen dataset and for ACES a second peak around 100 appears and grows. Of course, we are open to suggestions for improvement of the difficulty metric.
>
> > What does the shaded area represent on the graphs?
>
> They represent standard deviation, each experiment is repeated 3 times with different seeds.
>
> > OMNI-EPIC [1] utilizes a task archive [...]
>
> Of course! Thanks for pointing this out. We have added the reference.
>
> [1] Automatic Goal Generation for Reinforcement Learning Agents, Florensa et. al. 2017
>
> [2] Learning with AMIGo: Adversarially Motivated Intrinsic Goals, Campero et. al. 2020
>
> [3] Intrinsic Motivation and Automatic Curricula via Asymmetric Self-Play, Sukhbaatar et. al. 2017
>
> [4] Evaluating Large Language Models Trained on Code, Chen et. al. 2021
>
> [5] Language Models Can Teach Themselves to Program Better, Haluptzok et. al. 2022
>
> [6] Mathematical discoveries from program search with large language models, Romera-Paredes et. al. 2023
>
> [7] Large Language Monkeys: Scaling Inference Compute with Repeated Sampling, Brown et. al. 2024

---

> > ### Comment · Reviewer_ESVE · 2024-08-08
> >
> > Thank you for your detailed rebuttal. I found your answer and the main rebuttal PDF helpful.
> >
> > Overall, I think that the paper is making a modest, yet interesting contribution. Most importantly, the paper is technically solid and the claims are appropriately matched to the paper's contribution. In the end, the community would probably benefit and learn from this paper.
> >
> > Given these considerations, I am happy to increase my score.

---

### Official Review · Reviewer_NuHw · 2024-07-10

**Soundness:** 2
**Presentation:** 3
**Contribution:** 2
**Rating:** 4
**Confidence:** 4

**Summary:**

The authors propose Autotelic CodE Search (ACES) to generate programming puzzles, considering diversity and difficulty, borrowing ideas from intrinsic motivation and evolutionary search, relying on the assistance from LLMs for problem representation, skill label generation, diversity characterization, difficulty evaluation, and puzzle generation.

The authors conduct experiments to study quality of LLM-generated skill labels, and compare diversity and difficulty of the generated puzzles with existing methods and benchmarks.

**Strengths:**

+ generate diversified and challenging programming puzzles
+ borrow ideas from intrinsic motivation, in particular, the autotelic property, and evolutionary search, in particular, Map-Elites, an evolutionary quality-diversity (QD) method
+ conduct experiments studying quality of LLM-generated skill labels, and comparing with existing methods and benchmarks

**Weaknesses:**

- rely too much on LLMs
- follow a questionable approach
- overclaim

I am aware there are quite some papers following the framework of "self-generation, self-evaluation, and self-improvement, relying on one or more LLMs", some of them in the name of LLM-based agent or agentic AI.
This approach has a fundamental issue: no current LLM is perfect, so LLMs may make mistakes, and usually there is no mechanism to improve current LLMs in such framework.

As a result,  I vote to reject the paper, although it has merit, in particular, the idea for prompting diversity in generating programming puzzles and decent experimental results.

**Questions:**

Line 2,
"We propose a method that aims to automate this process by harnessing the power of state-of-the-art generative models to produce a diversity of challenging yet solvable problems, here in the context of Python programming puzzles."


The proposed approach relies on LLMs for problem representation, skill label generation, diversity characterization, difficulty evaluation, and puzzle generation.
The proposed approach does not provide a mechanism to improve the current imperfect LLMs, so that it does not close the loop.


Line 30,
"It would provide the necessary curriculum for open-ended learning machines [Colas et al., 2022] and may be a key component of automated scientific discovery [Grizou et al., 2020, Etcheverry, 2023]."

Similar to above, the feedback loop is not closed, so it is an over-claim.


This can be estimated by computing the empirical difficulty of a puzzle for a particular
39 solver: out of 50 attempts, the solver should solve the problem at least once (solvability) but as rarely
40 as possible (difficulty).


Line 44
"The standard approach for problem generation simply queries pretrained generative models with few-shot examples or specific instructions [Haluptzok et al., 2022, Honovich et al., 2023, Gunasekar et al., 2023]."

Why is it the standard approach? There are some papers working this way does not make it the standard way.


Line 134,
"Both datasets are pre-filtered to examples shorter than 1024 tokens to accommodate for limited context windows in LLMs."
This is an unnecessary limitation due to relying on LLMs.

Line 147,
"Puzzles for which no solution is found are considered invalid and are discarded."
This is problematic, at least, not ideal.
Line 202,
"If the puzzle is not solved by the solver model in 50 attempts, it is considered unsolvable and discarded."
It is the limitation of current LLMs, and the authors do not propose a mechanism to overcome it.

Line 150,
"this difficulty-based metric measures a ground-truth objective: how hard a given puzzle is for a target LLM solver"
It is not ground-truth: LLM solvers are not perfect, so some measure wrt an LLM is not a ground-truth.


Line 151,
"Our experiments found that this difficulty measure mostly transfers across models: a puzzle that is harder for one model is often harder for others (see Section 4.4)."

This may not say much about transferability, since most LLM models were built with similar methods.


Line 164,
"We sample our 20 tags from a larger list generated by an LLM (GPT-4-0125)"

Use an imperfect LLM, namely, GPT-4, as reference has inherent limitations.


Line 194,
"to drive discovery of puzzles with novel skill combinations, we rely on the LLM recombining elements from puzzles close to the target cell along with very often selecting target cells without any representatives."
Again, limitations of current LLMs.


Line 335,
"We expect that progress along the three lines can be made by harnessing the increasing capabilities of future language models and will automatically translate into higher performing autotelic generative models of problems."
It is not reasonable to postpone the development of key components of a proposed approach to something the authors do not have control of, in particular, "future language models".


Line 354,
"In artificial intelligence, one could envision a self-play loop where a learning agent iteratively generates a diversity of problems maximizing its current learning progress (quality metric), then trains on them to augment its competence [Sukhbaatar et al., 2017, Silver et al., 2017, Colas et al., 2022]."
This is an overclaim and it is misleading.
It replies on imperfect LLMs and do not attempt to improve them.

**Limitations:**

The authors do not discuss enough the limitations of reliance on imperfect LLMs, and do not propose to improve imperfect LLMs.

---

> ### Author Rebuttal · Authors · 2024-08-06
>
> We thank the reviewer for their comments and hope to be able to convince them that our assumptions are sound.
>
> Reviewer NuHw seems to reject the possibility of leveraging LLMs in any kind of systems, based on the fact that LLMs are sometimes wrong. That LLMs are not perfect at any of the tasks we use them for is of course true, but that does not prevent us from using them in a useful way (here to build diverse sets of puzzles to evaluate itself and other models).
>
> For each of the components we use LLMs for, they clearly outperform other available options:
>
> * For puzzle generation, other program synthesis methods would hardly give interesting, human-relevant programming puzzles;
> * For measuring diversity (creating niches for our QD algorithm), we tried using voronoi tessellations as is usually done in high-dimensional Map-Elites [1], but it works less well than the skill labels using the LLM (compare ELM-CVT with ELM, Figure 3) and it is less interpretable. The skill labeling performance itself is not so bad, see responses to reviewer qqHU, and section 4.2.
> * For measuring difficulty we need to use LLMs as solvers, since that’s the measure of difficulty we want to measure (and it is linked to pass@k, the measure of success on coding datasets);
>
> In addition to these considerations, we add checks to ensure our generated puzzles make sense (rejecting puzzles with no solution, see also Section 3.3). We use several measures of diversity, and difficulty with respect to different models, to make sure our evaluations are sound. Can the reviewer provide a more specific description of LLM limitations that would make our approach unsound? What alternative could be considered to implement the puzzle generation, measure diversity or difficulty here?
>
> > overclaim
>
> The reviewer cites excerpts of our motivation and conclusion and seems to consider we claim these as parts of our contribution, which they are not. Our contributions are listed in the last paragraph of the intro, at the end of the abstract, and at the beginning of our conclusion.
>
> ### Questions
>
> We reply to the reviewer’s comments here, but some of them are not really questions. It would be helpful and constructive for the reviewer to be more specific in their remarks; often the perfectness of LLMs is attacked, and we agree these models are far from perfect; however if alternative methods or fields or studies were mentioned it would help us write more targeted responses.
>
> > The proposed approach does not provide a mechanism to improve the current imperfect LLMs, so that it does not close the loop.
>
> We do not propose to directly improve LLMs in this paper; not every paper needs to directly improve LLM state of the art. We simply present methods for generating diverse and challenging datasets, and argue that they could be used to evaluate LLMs in the future. Evaluating LLMs on challenging datasets is part of the process leading to their improvement. (and we confirm by the way that combining evolutionary methods with LLMs is fruitful, as was demonstrated in FunSearch [2] or QDAIF [3] among others). Our additional experiments demonstrated that our method significantly enhances the performance of current "imperfect LLMs." The LLM fine-tuned on our approach-generated archive outperformed the base model by a substantial margin, even surpassing larger models.
>
> > Similar to above, the feedback loop is not closed, so it is an over-claim.
>
> This is part of the motivation of the paper (beginning of the introduction), not its contributions.
>
> > Why is it the standard approach? There are some papers working this way does not make it the standard way.
>
> It is the standard approach insofar few-shot prompting is a common method for generating domain-specific data with LLMs. If you have another idea/method in mind, we would be interested in hearing about it.
>
> > [on discarding unsolvable puzzles] This is problematic, at least, not ideal.
>
> We agree that this is less than ideal (see also our response to reviewer ESVE) but we have no principled method for determining whether a puzzle is solvable without generating a solution ourselves. We are of course very open to additional suggestions here.
>
> > This may not say much about transferability, since most LLM models were built with similar methods.
>
> We report an observation here. This is also confirmed in our new experiments. As for non-LLM solvers, we do not know but consider it interesting to correlate difficulties of puzzles across solver types (and humans).
>
> > [on generating the initial list of skills] Use an imperfect LLM, namely, GPT-4, as reference has inherent limitations.
>
> As always we agree. That is why we filter the set afterwards to make sure the skills make sense. The alternative is to generate the list ourselves, based on puzzle categories in algorithm books and programming contests. In practice both methods (we tried both) led to very similar lists.
>
> > "to drive discovery of puzzles with novel skill combinations, we rely on the LLM recombining [...]" Again, limitations of current LLMs.
>
> Could you be more specific?
>
> > It is not reasonable to postpone the development [...]
>
> This is not what we meant in L338 (conclusion). We stated that the approach probably scales as the LLMs become better (this is a good property of the method). We confirm this in our new experiments.
>
> > "In artificial intelligence, one could envision a self-play loop [...]” This is an overclaim and it is misleading. It relies on imperfect LLMs and do not attempt to improve them.
>
> This is one of our last paragraphs and is future work/discussion! We do not claim to instantiate a self-play loop in this paper.
>
> [1] Using Centroidal Voronoi Tessellations to Scale Up the Multi-dimensional Archive of Phenotypic Elites Algorithm, Vassiliades et. al. 2016
>
> [2] Mathematical discoveries from program search with large language models, Romera-Paredes et. al. 2023
>
> [3] Quality-Diversity Through AI Feedback, Bradley et. al. 2023

---

> ### Comment · Reviewer_NuHw · 2024-08-09
>
> Thanks the authors for the detail rebuttal.
> I admit there are merits in the submission, and I am aware there are many papers similar wrt (unreliable) LLM-based methods.
> However, I am not convinced.
>
>
> There are two fundamental issues with the submission:
> 1) fully rely on an imperfect LLM;
> 2) rely on problematic method (evaluation of pass@k).
>
>
> 1)
>
> As in my original review:
> I am aware there are quite some papers following the framework of "self-generation, self-evaluation, and self-improvement, relying on one or more LLMs", some of them in the name of LLM-based agent or agentic AI.
> This approach has a fundamental issue and is misleading: no current LLM is perfect, so LLMs may make mistakes, and usually there is no mechanism to improve current LLMs in such framework.
>
>
> Line 247
> "Puzzle generation, solution generation, description generation, and puzzle labeling are all implemented with the state-of-the-art open source model Llama 3 70B, quantized in 4 bits, with a temperature parameter of 0.8."
>
> All these components are based on an imperfect LLM, so all of them are not fully reliable.
>
> 2)
>
> How to evaluate "pass" (in pass@k)?
> How to guarantee the correctness of the evaluation?
>
> In the current popular approach, like HumanEval, if a solution passes several test cases, then it is a pass.
> This is problematic: Testing only can not guarantee the correctness of a piece of code. Formal verification is required.
>
> This is not a reliable evaluation.
>
> As a result: no guarantee of the quality of generated puzzles, some of them may be wrong.
>
> A question from reviewer qqHU:
> "How does the author ensure the correctness of the generated test program $f$"?
>
> Can authors ensure the correctness of the generated test program?
> The binary answer should be NO.
> With LLMs, it is best effort.
>
>
> From authors' general rebuttal:
> "Reviewer NuHw rejects the approach on the basis that it is built on LLMs, but we note that generating challenging evaluation sets is an instrumental application to create better LLMs in the future."
> This is both true and false. In a top AI venue like NeurIPS, we want to see papers with reliable method / results, without fundamental issues. The current submission has a fundamental issue: reliance on an imperfect LLM.
>
> FunSearch and AlphaGeometry are different: they have reliable verifier in the loop.
>
> This is a problem for the community, in particular, those working on LLM-based method.
> Top AI venues like NeurIPS should not publish more such papers without fully reliable methods. This is misleading.

---

> ### Author Response · Authors · 2024-08-12
>
> We thank the reviewer for their reply and we are happy they find “there are merits in the submission”. Their first concern is on building evaluations using LLMs to generate, label, and compute difficulty since LLMs can make mistakes. Their other concern seems to be on the computation of the pass@k. Reviewer qqHU shared part of the same concerns and we also refer to our response to them for additional discussion on the validity of using LLM-generated puzzles to evaluate LLMs themselves.
>
> > All these components are based on an imperfect LLM, so all of them are not fully reliable.
>
> We would like to point out that synthetic data is successfully used for supervised fine-tuning to enhance current LLM capabilities, even if LLMs can make mistakes. This is a common part of modern LLM post-training. For instance, in the Llama 3.1 paper [1] (Section 4.3.1) the authors use a previously trained version on llama 3 405B to generate coding exercises (based on random code snippets from “various sources”) and use execution feedback to ensure the generated solutions are correct according to the provided unit tests. They cannot fully guarantee the accuracy of generated solutions but they show this synthetic data is helpful for increasing model capabilities, illustrating the usefulness of the method.
> As explained in our reply to Reviewer qqHU, synthetic benchmarks and evaluations (beyond using GPT4 as a proxy for human evaluation) have also been useful to uncover surprising or harmful model behavior [2, 3, 4].
>
> > rely on problematic method (evaluation of pass@k). HumanEval [...] is problematic [...] Formal verification is required.
>
> On HumanEval: You are right that, short of formal verification one cannot verify that a solution solves exactly the problem as it is formulated in the problem description. However this is the same situation as all coding competitions and evaluations directed at humans, where one relies on passing test cases to judge the validity of the solution (see LeetCode evaluations for instance), as well as most of the software written everyday in industrial applications (except some critical domains like aeronautics where software is verified). You seem to suggest that all coding competition grading is flawed, as well as all code model evaluations (for which HumanEval has been the main inspiration). While the evaluations are not perfect, this has consistently guided progress with coding LLMs. Would you similarly reject all papers using HumanEval to evaluate a code model on the grounds that the solutions generated by the models might be flawed?
> Formally verifying solutions to general programming problems (using Coq, Lean, Isabelle, etc) is far from a standard in the field of code LLMs and turning arbitrary programming puzzles into a formal specification is out of reach for current autoformalization methods (and out of the scope of this paper, this is a research field in itself).
>
> > Can authors ensure the correctness of the generated test program?
>
> We are in a different setup from HumanEval. By definition of a P3 puzzle, if `f(g()) is True` (according to the Python interpreter, see also Section 3.1 and [5]), the solution is correct. If at least one solution is found, the puzzle is solvable (thus correct). This does not guarantee the puzzle is interesting but the P3 correctness criterion is straightforward to judge. We increase the chance the puzzles are interesting by optimizing how hard (for LLMs) and diverse they are.
>
> > FunSearch and AlphaGeometry are different
>
> Funsearch does not generate problems so is not directly comparable. AlphaGeometry does generate geometry problems, and uses a specialized geometry DSL to ensure correctness of solutions. This is the equivalent to the Python interpreter in our case.
>
> We hope this discussion answers some of your concerns and we thank you for the opportunity to improve the paper.
>
> [1] The Llama 3 Herd of Models, Llama team 2024
>
> [2] Discovering Language Model Behaviors with Model-Written Evaluations, Perez et. al. 2022a
>
> [3] Red Teaming Language Models with Language Models, Perez et. al. 2022b
>
> [4] Rainbow Teaming: Open-Ended Generation of Diverse Adversarial Prompts, Samvelyan et. al. 2024
>
> [5] Programming Puzzles, Schuster et. al. 2021

---

> > ### Comment · Reviewer_NuHw · 2024-08-13
> >
> > 1.
> > “We would like to point out that synthetic data is successfully used for supervised fine-tuning to enhance current LLM capabilities, even if LLMs can make mistakes.”
> >
> > This means the method is dubious.
> >
> > A top AI venue like NeurIPS should not accept a paper with a dubious method.
> >
> > 2.
> > "On HumanEval: You are right that, short of formal verification one cannot verify that a solution solves exactly the problem as it is formulated in the problem description."
> > "Would you similarly reject all papers using HumanEval to evaluate a code model on the grounds that the solutions generated by the models might be flawed?"
> >
> > The right question you should ask is: Why have such papers been accepted?
> >
> > A top AI venue like NeurIPS should not accept a paper with a flawed method.
> >
> > 3.
> >
> > "AlphaGeometry does generate geometry problems, and uses a specialized geometry DSL to ensure correctness of solutions. This is the equivalent to the Python interpreter in our case."
> >
> > AlphaGeometry can guarantee the correctness of the proof.
> > Your method can not guarantee the correctness of the code.
> >
> > A top AI venue like NeurIPS should not accept a paper with a fundamental flaw.

---

### Official Review · Reviewer_qqHU · 2024-07-11

**Soundness:** 2
**Presentation:** 2
**Contribution:** 3
**Rating:** 6
**Confidence:** 4

**Summary:**

This paper proposes Autotelic CodE Search (ACES), a method to generate diverse and challenging Python programming puzzles using state-of-the-art generative models. ACES optimizes for both the diversity and difficulty of problems by representing them with semantic descriptors that describe the programming skills required to solve them and measuring difficulty empirically with the pass rate in 50 tries. The method iteratively prompts a large language model to create new puzzles, using previously generated related ones as few-shot examples. ACES outperforms baseline methods, producing puzzles that are more diverse and three times more challenging than existing Python benchmarks, paving the way for automated curriculum learning and scientific discovery.

**Strengths:**

1. The method is reasonable and the experiment results are promising, showing the advantage of this method.
2. The method proposed appears to be a general approach that can be extended to fields beyond code generation, as long as semantic descriptors can be set.
3. I believe that automatically generating benchmarks to test the shortcomings of large language models is a meaningful research direction.

**Weaknesses:**

1. How do the results differ when using different models as solvers? In this paper, the difficulty is measured by how easily tasks are passed by the Llama-3-70b model, making Llama-3-70b an extremely important model for evaluating the capabilities of other models. Any model that does not perform consistently with Llama-3-70b is considered unable to pass certain tasks. Although the authors claim that "There is a clear correlation between solver performance across different problem sets, which indicates that tailoring problem generation to a specific model (Llama-3-70b) generates problems that are challenging for other models too,", it seems that this is actually just an assumption. In other words, the authors cannot explain whether the top left corner in Figure 4 (b) is overfitted to HumanEval or simply inconsistent with the Llama-3-70b model. One way to solve this is to present results on more solver models.
2. How does the author ensure the correctness of the generated test program $f$ (i.e., that the test program and the natural language description of the puzzle are consistent)? A problem can become very difficult if $f$ and the natural language description are inconsistent, and manual checking is challenging to cover all 6400 problems.
3. Some parts are not clear. (1) I'm confused about which one is your final method, ACES or ACES-ELM? It seems that ACES-ELM is an ablation of ACES (Section 3.4) but it is taken as the main method in experiments like  Figure 4(b). (2) In section 4.2, what do these numbers (F1 scores) represent and which method should they be compared with? I think adding a baseline would help explain this better.
4. This paper evaluates the quality of the benchmark based on diversity and difficulty (which is actually the passing rate of the LLM, rather than an objective difficulty). However, I believe that difficulty is just one of the criteria for evaluating a benchmark, and this approach is somewhat one-sided. Other criteria, such as whether the benchmark meets practical usage needs (like HumanEval, which is manually designed to ensure that these programming tasks do not appear in the training set and are commonly used in real scenarios), or whether the benchmark covers a broader range of scenarios (like DS-1000, which targets scientific computing scenarios), are also important evaluation standards.

**Questions:**

1. Why there are 21700 niches in total? If we combine these tags in various ways, we should be able to come up with $2^{20}-1$ niches.
2. Some typos: Tesla V100 SXM2 32 Go (GB?), Appendix Section! A.2.
3. Can the authors test the generated tasks on closed-source LLMs like GPT (it is okay to say no)?

**Limitations:**

The authors have discussed the limitations well.

---

> ### Author Rebuttal · Authors · 2024-08-06
>
> We thank Reviewer qqHU for their review. We thank the reviewer for pointing out the experiments are promising, the research direction interesting, and for noting the method is general and applicable to other domains beyond code. We hope to address the reviewer’s concerns.
>
> > How do the results differ when using different models as solvers? [...]
>
> We have performed additional experiments with Llama 405B and Mistral Large 2. For each new experiment, the LLM is used to generate puzzles, label skills, and to measure difficulty. See the pdf attached to the main response, notably Figure 1 (a), (b) and (c). The results present evidence of transfer of puzzle difficulty across models: models that have low scores on our datasets generated with Llama 70B also have low scores on the datasets generated by Llama 405B and Mistral, as well as on LiveCodeBench [1] (specifically built to avoid contamination) and BigCodeBench [2]. Figure 1 (a) shows that only HumanEval escapes this correlation, suggesting high levels of contamination or models (especially the small ones) optimized to perform well on that benchmark. We have plotted the correlation between scores on all datasets (averaged over models) in Figure 1 (b). The correlation between all dataset scores, save for HumanEval, is clearly visible. We hope this answers your concerns on overly relying on a single solver. As additional evidence, we find good correlation between model size and overall performance on our datasets as well as LiveCodeBench and BigBench, which is what one would expect if coding capability was monotonically and smoothly related to model size.
>
> > How does the author ensure the correctness of the generated test program [...]
>
> We agree that it is not feasible to manually ensure consistency of the generated puzzles.
> For now we limit ourselves to ensuring the puzzle has at least one solution, and we generate the description from the puzzle and the solution using a specialized prompt. To increase puzzle-description consistency, we could additionally use a critic LLM, or use human labels as guidance. We also note that for the original P3 dataset, usually the description of puzzles are not given to models (see here https://github.com/microsoft/PythonProgrammingPuzzles/blob/main/notebooks/Intro.ipynb).
>
> > Some parts are not clear. (1) I'm confused about which one is your final method, ACES or ACES-ELM?
>
> ACES-ELM is a goal-directed quality-diversity method (as in ACES, we target a cell and instruct the model/choose the examples to be able to reach it), thus not really an ablation. We agree that the presentation could be better (we presented it after ELM because it uses the same mutation mechanism). Because they are goal-directed methods and exhibit similar performance we consider both methods to be our final methods. We changed section 3.3 and 3.4, where we present ACES and ablations respectively, to reflect this.
>
> > In section 4.2, what do these numbers (F1 scores) represent [...]?
>
> These correspond to skill labeling performance compared with a human-labeled ground truth, considered as a per-skill binary classification process. More precisely, for each puzzle, we have a set of ground-truth human labels. This gives us a classification problem with binary labels (is present, not present) for each skill index. To provide a random classification baseline one could do the following: for each of the 60 puzzle instances we manually assigned labels to, randomly assign 1 to 5 skills and compute the precision $p_0$, the recall $r_0$ and the f1 $f_0$. With 20 random draws of this random classifier we get $p_0 = 0.13$, $p_0 = 0.14$ and $f_0 = 0.12$ on average; well below the LLM judge’s performance. We added a footnote with this result.
>
> > [difficulty] which is actually the passing rate of the LLM, rather than an objective difficulty
>
> Unfortunately we cannot be much more objective than this, and this measure of difficulty is related, in any case, to the metrics the community uses to evaluate code models. We could average on a family of LLM to get an estimate of the difficulty of a puzzle, for LLMs in general. If we want to get difficulty estimates for humans, which might be quite different, there is unfortunately little practical way to collect a large scale dataset to e.g. train a classifier. Any idea you would have is of course welcome!
>
> > However, I believe that difficulty is just one of the criteria for evaluating a benchmark, and this approach is somewhat one-sided.
>
> We agree! We realize that there is much more than difficulty in a good puzzle. HumanEval was a well-crafted dataset and large amounts of human labor was needed to make a benchmark of this quality. It is now commonly accepted that HumanEval is present in the LLM’s train sets, making the results on this benchmark potentially unreliable. We believe that in the future, synthetic benchmarks will be an important part of LLM evaluation (but not all of it), in the spirit of how chatbot competitions and GPT4-judged responses are used to inform our current evaluations of LLMs. With this work we make a step in this direction, by showing that leveraging insights from the evolutionary computing and intrinsic motivation literature we can optimize for datasets with low pass rate and coverage in a predefined skill space.
>
> ### Questions
>
> > Why there are 21700 niches in total?
>
> We limit to 5 skills max, to avoid targeting puzzles with unrealistic skill combinations.  ($\sum_{k=0}^{5} \binom{20}{k} = 21700$) We forgot to report this detail, we have added it in section 3.2.
>
> > Can the authors test the generated tasks on closed-source LLMs like GPT (it is okay to say no)?
>
> See our results with large open models.
>
> [1] LiveCodeBench: Holistic and Contamination Free Evaluation of Large Language Models for Code, Jain et. al. 2024
>
> [2] BigCodeBench: Benchmarking Code Generation with Diverse Function Calls and Complex Instructions, Zhuo et. al 2024

---

> > ### Comment · Reviewer_qqHU · 2024-08-10
> >
> > Thanks for the response! Many of my concerns have been addressed and for other ones, although the authors did not provide a clear solution and acknowledged this as a limitation of their work, I believe that overall, this work represents a commendable attempt.
> >
> > Now, I think there is only one question that remains unanswered:
> >
> > As I mentioned earlier, and as reviewer NuHw also noted, how can puzzles generated by LLMs, which cannot guarantee correctness, serve as benchmarks to evaluate LLMs themselves (or other LLMs)?
> >
> > I know the following information, but I don't think this answers the question.
> >
> > 1. I know that some benchmarks use LLMs (mostly GPT-4) as the evaluation metric because these benchmarks require human evaluation, which cannot be automated. GPT-4 here serves as an approximate automated evaluation method.
> > 2. I know that LLMs can automatically generate training datasets for training purposes because we have golden benchmarks to evaluate the final training outcomes. However, if we use LLM-generated benchmarks for the final evaluation, there will be nothing to ensure their performance.
> > 3. Perhaps we can use benchmarks generated by stronger LLMs to evaluate weaker ones, but how do we evaluate the stronger LLMs? If we use human-designed benchmarks to evaluate stronger LLMs, then why not use them directly to evaluate weaker ones?
> >
> > This is quite important and I hope the authors can discuss this question further.

---

> > > ### Author Response · Authors · 2024-08-12
> > >
> > > We appreciate your thoughtful feedback and the opportunity to address this important question. We understand your concern about using puzzles generated by LLMs as benchmarks to evaluate LLMs themselves. We'd like to offer the following perspectives on this matter:
> > >
> > > * **Objective solvability**: Since a P3 puzzle is defined by its test case only (and the language description is generated afterwards to provide help), puzzles with at least one solution are by definition correct (i.e. solvable). This ensures a baseline level of validity for the benchmarks.
> > >
> > > * **Puzzle quality and interest**: Although all solvable puzzles are correct, this doesn’t mean they are interesting or worthwhile. We acknowledge that evaluating puzzles 'interestingness' or relevance is subjective and challenging to automate. Our approach with semantic labels provides users some control over the generation process, allowing them to orient puzzles towards abstract features they deem important. While we cannot fully automate the assessment of puzzle quality, our method generates problems that are more difficult and diverse in customizable ways, potentially increasing their overall quality and relevance. We could also add an interesting metric judge by an LLM [1, 2] and/or an educational value [3, 4] to filter the least interesting puzzles.
> > >
> > > * **Preexisting automated benchmarks**: we would like to point out that synthetic benchmarks have already led to discovery of interesting features of LLMs [5, 6, 7]
> > >
> > > * **Human-in-the-loop approach**: Our method significantly speeds up benchmark generation by producing a pool of potentially interesting puzzles from which humans can select, in a similar fashion as BigCodeBench or HumanEval Plus where the authors used a human in the loop method to guide or filter LLM generated data;
> > >
> > > * **Potential for automated quality assessment**: In the future, we could explore using preference models or tournament-style competitions with the top k LLMs to automatically select top puzzles from the generated set. This approach has shown promise in other areas of AI evaluation.
> > >
> > > * **Comparative evaluation**: To address the concern of using LLM-generated benchmarks to evaluate LLMs, we could compare LLM performance with human performance on these benchmarks. This approach has been used effectively in other studies comparing the reasoning capabilities of LLMs and humans [8]. Additionally, we note that model scores on our generated sets and on LiveCodeBench (composed of newly released coding puzzles to avoid contamination) are strongly correlated (See additional Figure 1c). We take this as evidence that the generated set measures underlying coding capabilities;
> > >
> > > > Perhaps we can use benchmarks generated by stronger LLMs to evaluate weaker ones, but how do we evaluate the stronger LLMs?
> > >
> > > In ACES, increasing the number of attempts leads to harder problems (since we can optimize for puzzles solved once out of N with N being large). Thus, even weaker LLM can generate difficult tasks for better LLMs, as shown in our additional experiment where Llama-3.1-405B only gets 74% pass@1 on a dataset generated with ACES-ELM with Llama-3-70B (Additional Figure 1a). This is directly due to the fact that ACES can optimize for empirical difficulty.
> > >
> > > * Although we have shown that puzzle difficulties transfer between Llama and Mistral (additional Figure 1a, Mistral Large and Llama 3 405B have similar scores on ACES-ELM datasets generated by both models) we could also take the three or four best LLMs and generate multiple archives with our method, then merge them to get a test set; as each LLM has some strengths and weaknesses, the resulting archive should be balanced in terms of puzzle difficulties for other models.
> > >
> > > We hope this addresses your concern and provides evidence for the usefulness of synthetically-generated evaluation sets. Given that many of your previous concerns have been addressed and you've acknowledged this work as a commendable attempt, we kindly request that you consider updating your score to reflect your opinion of the paper.
> > >
> > > Thank you again for your valuable feedback and the opportunity to improve our work.
> > >
> > > [1] Omni: Open-endedness via models of human notions of interestingness, Zhang et. al. 2023
> > >
> > > [2] OMNI-EPIC: Open-endedness via Models of human Notions of Interestingness with Environments Programmed in Code, Faldor et. al. 2024
> > >
> > > [3] Cosmopedia: how to create large-scale synthetic data for pre-training, Ben Allal et. al. 2024
> > >
> > > [4] Textbooks Are All You Need, Gunasekar et. al. 2023
> > >
> > > [5] Discovering Language Model Behaviors with Model-Written Evaluations, Perez et. al. 2022a
> > >
> > > [6] Red Teaming Language Models with Language Models, Perez et. al. 2022b
> > >
> > > [7] Rainbow Teaming: Open-Ended Generation of Diverse Adversarial Prompts, Samvelyan et. al. 2024
> > >
> > > [8] Beyond the imitation game: Quantifying and extrapolating the capabilities of language models, Srivastava et. al. 2022

---

> > > > ### Comment · Reviewer_qqHU · 2024-08-13
> > > >
> > > > Thanks for the response! I'm satisfied with the response except for the "Potential for automated quality assessment" part (how could we do that?). And the result that even weaker LLM can generate difficult tasks for better LLMs is impressive.
> > > >
> > > > I updated my score to 6.

---

> > > > > ### Author Response · Authors · 2024-08-13
> > > > >
> > > > > Many thanks for updating your score!
> > > > >
> > > > > On automated quality assessment one could use a model trained on human preference data over puzzles (in a similar way to reward models in RLHF), or use a classifier trained on sets of (good quality, bad quality) puzzles (in a similar way as classifiers used to filter pretraining data for large-scale LLM training).

---

### Official Review · Reviewer_LkwK · 2024-07-13

**Soundness:** 3
**Presentation:** 2
**Contribution:** 3
**Rating:** 6
**Confidence:** 4

**Summary:**

In this paper, the authors propose a method to automate the generation of challenging coding puzzles. By leveraging quality-diversity (QD) algorithms and defining novel metrics, their approach produces coding puzzles that are more diverse and challenging than existing benchmarks.

**Strengths:**

- The proposed algorithms and metrics have the potential to be significant and impactful. Generating diverse learning environments and measuring their diversity has been a major challenge for the community. The methods proposed in this paper represent valuable attempts to address these important issues.
- The validation of the method is solid. For example, the quality of skill descriptors generated by an LLM is evaluated with human assessments, and extensive baselines and model variants are implemented and compared to demonstrate the effectiveness of the proposed method.
- The proposed method generates more diverse and challenging coding puzzles, which could be highly beneficial for future studies on LLM coding agents.

**Weaknesses:**

### The presentation could be improved:
- The authors might consider explaining QD and Map-Elites more clearly. While I am familiar with these methods and did not have difficulty, readers who are not as familiar with these topics might find them challenging to understand.
- As someone familiar with related topics, I found the proposed method difficult to understand. My current understanding is that the difficulty metric is used as fitness, and two diversity metrics are two dimensions in the Map-Elites archive. Is this correct? Regarding embedding diversity, is it the higher the better (since it's more diverse)? If that is the case, does it make sense to use it as one dimension in the Map-Elites archive?
- The authors might consider introducing the motivation behind their design choices before explaining them (e.g., P4, L128; the three metrics in Section 3.2). Additionally, for the paragraph at P5, L185, I don't fully understand how the design reflects the intuition.
- There are some questions in the question section that I do not fully understand.

**Questions:**

- How the target cell get selected?
- I don't fully understand the motivation behind the ablation using ELM with semantic categories. What does "ablation of goal-directedness" mean?
- Since the approach did not include the training of LLMs, do the authors consider using more powerful LLMs like GPT-4 or Claude-3.5?
- Which variant is the final proposed method? Is it ACES-ELM, ACES, or another variant?
- Using visualization tools like t-SNE could be beneficial to show the distribution of generated coding challenges. It will be interesting to see the clustering of different skills being tested in these generated problems.
- Figure 3(f) seems counterintuitive to me. Do the authors have any insights into why the proposed method generates either the most challenging tasks or very easy ones? What happens to the intermediate tasks? Is it the case that the model cannot generate them, or are they rejected after generation?

**Limitations:**

Limitations are addressed in the Discussion section

---

> ### Author Rebuttal · Authors · 2024-08-06
>
> We thank Reviewer LkwK for their thoughtful comments, and are glad they find the problem we study important, and the evaluation solid. We now aim to clarify some points in our response.
>
> > The presentation could be improved.
>
> We thank you for this feedback. We have modified section 3.4 by being more precise when introducing Map-Elites.
>
> > My current understanding is that the difficulty metric is used as fitness, and two diversity metrics are two dimensions in the Map-Elites archive.
>
> That is not quite correct, thanks for pointing out that the explanation is unclear. Fitness is the quality metric, and the archive is composed of cells corresponding to a maximum of 5 discrete programming skills (among 20).
>
> > Regarding embedding diversity, [...] does it make sense to use it as one dimension in the Map-Elites archive?
>
> We have two diversity measures: skill (or semantic) diversity, which is the number of unique skill combinations present in the generated set, and embedding diversity, which is the average pairwise distance between puzzles in an embedding space. These measures are NOT dimensions of the archive. Embedding diversity, which is independent from semantic diversity (and computed with different models than the one used for generation/skill labeling), is used as an independent measure of diversity (since skill labeling is imperfect). The measures agree (see Figure 3).
>
> > The authors might consider introducing the motivation behind their design choices before explaining them.
>
> Thank you for your suggestion! The reason for using P3 is twofold:
> * Code LLMs are a focus of the ML community, and for good reason. However, state-of-the-art models are getting so good as to mostly saturate today’s programming benchmarks (see also responses to Reviewer qqHU). Applying QD search with difficulty as a fitness metric is of obvious use there. We manage to create coding datasets that are challenging for current capable LLMs;
> * P3 is a relatively simple, language-based, open-ended domain where one can check the solutions exactly to compute how hard a task is (contrary to the tasks used in QDAIF [1]).
>
> As for the metrics introduced in section 3,2:
> * **Difficulty**: creating challenging problems was our aim and is important. The reasons for the precise formulation are given in the response to reviewer ESVE, third response to questions.
> * **Skill diversity**: These dimensions of variation across puzzles are intuitive for humans and are useful for downstream applications (benchmark generation, and in possible future work, generation of programming exercises for students).
> * **Embedding diversity**: see previous response.
>
> ### Questions:
>
> > How is the target cell selected?
>
> In ACES (goal-reaching method), we select a cell uniformly at random among all possible cells, filled or otherwise (Section 3.3, second paragraph, as well as Figure 1). In ELM (mutation-based method), we select a cell uniformly at random among the ones that are filled (Section 3.4, first paragraph);
>
> > I don't fully understand the motivation behind the ablation using ELM with semantic categories. What does "ablation of goal-directedness" mean?
>
> ELM selects an existing solution and applies an undirected mutation to it using the LLM-based mutation operator. ELM with semantic categories extends ELM by storing existing solutions in the skill-based archive introduced in our paper – and we show this improves performance compared to ELM using Voronoi cells. (It is also very similar to QDAIF). ACES replaces the undirected mutation operator by a goal-directed mutation: it first selects a target cell (empty or filled), then tries to fill it by prompting the LLM to generate a new problem characterized by the skills of the target cell. ACES-ELM also selects a target cell, but tries to fill it by mutating an existing puzzle; it is a goal-directed version of ELM. Our hypothesis was that goal-directed mutation is better than undirected mutation. Our hypothesis is confirmed in our experiments as niche coverage grows faster with goal-directed variants (ACES-ELM and ACES) than with non-directed variants (ELM, ELM-CVT). Additionally, the goal-directed variants also produce higher-difficulty puzzles: this might be due to the higher diversity of puzzles which allows the algorithm to find higher-difficulty puzzles as well, as is often the case in QD search.
>
> > [...] Do the authors consider using more powerful LLMs like GPT-4 or Claude-3.5?
>
> The experiments are somewhat costly to run with the more expensive closed-source API models. We did experiments with the new Llama model and Mistral large 2 that are GPT-4 level (see general reply). See general reply for results.
>
> > Which variant is the final proposed method?
>
> Both methods exhibit minor variations and yield comparable results. Given that ACES-ELM demonstrates a slight edge in diversity and difficulty, we recommend it as the preferred variation of the method. We have updated section 3.3, and section 3.4, to reflect this.
>
> > Figure 3(f) seems counterintuitive to me. Do the authors have any insights into why the proposed method generates either the most challenging tasks or very easy ones?
>
> We are not sure. In the beginning of training the only puzzles generated are the easy ones (difficulty close to 0); as generation progresses with ACES, a larger and larger proportion of puzzles are close to maximum difficulty (See new results, Figure 1 (e)). We instruct the model to generate only puzzles of the highest difficulty, by including difficulty scores of the few-shot examples in the prompt. One possibility is that this works so well that it heavily skews the distribution of difficulties. Another explanation could be that we heavily skew sampling of example puzzles based on their difficulty score, leading to creating of puzzles with similar difficulties.
>
> [1] Quality-Diversity Through AI Feedback, Bradley et. al. 2023

---

> > ### Comment · Reviewer_LkwK · 2024-08-09
> >
> > Thank you for thoroughly addressing my comments and providing the additional results. After carefully reviewing your response, I find that my primary concerns and points of confusion have been addressed. I support the publication of this paper in NeurIPS and will maintain my current score.

---

### Author Rebuttal · Authors · 2024-08-06

We thank all reviewers for their reviews and the time they spent reading the paper, and we hope this rebuttal and the discussion period will be productive, answer questions and overall lead to a better paper.

There are two ways to read this paper, depending on which background one has. The takeaways from the paper will be slightly different in each case.

* **For general machine learning readers with an interest in (code) LLMs**, the contribution of this paper is showing that one can use a measure of puzzle difficulty – as well as LLM judgements of which programming skills are required to solve a puzzle – to create sets of hard and diverse programming problems. We envision this as a step to creating new programming benchmarks for code LLMs. Reviewer ESVE worries about the application of the method, but we think the contribution to generating difficult synthetic benchmarks is quite clear, as Reviewer qqHU also notes in his comments. Reviewer NuHw rejects the approach on the basis that it is built on LLMs, but we note that generating challenging evaluation sets is an instrumental application to create better LLMs in the future.

* **For readers interested in open-ended search**, we present algorithmic improvement over LLM-augmented map-elites (ELM), the baseline method that is extensively used in other papers [1, 2, 3]. We build a goal-directed algorithm that explicitly targets potentially empty cells, instead of using random mutation like Map-Elites-inspired approaches, and we demonstrate that this is helpful. Importantly, the goal-directed property of ACES comes from 1) the few-shot abilities of LLMs, to create a puzzle that resembles puzzles chosen near the target cell and 2) their instruction-following abilities, to create a puzzle in the desired cell (with the right skill combination). The goal-directed evolutionary algorithm underlying ACES could not be implemented without LLMs; this algorithmic improvement takes full advantage of the use of instruction-following LLMs, with few-shot abilities, as mutation operators.

We present a series of complementary results in the attached pdf.

### Results with different models
To respond to Reviewer qqHU’s concerns of over-reliance on a single model, and to calls by Reviewers LkwK, qqHU to use bigger, state-of the art models, we performed additional experiments with the new Llama 405B model and Mistral large 2, models on par with GPT-4. Using those LLMs leads to a better Quality-Diversity score overall, up to 25.6% for Mistral Large and 12.3% for Llama 405B (using Llama 405B and Mistral large 2 both for the difficulty metric and skill labeling). This demonstrates how ACES scales with models of higher size. Evaluating Mistral Large on the dataset generated by Llama 405B, as well as the other way around, we find pass@1 scores of 56.7% and 58% respectively. This demonstrates the effectiveness of our method in generating a challenging benchmark, even for state-of-the-art models, as well as the fact that difficulty measures transfer across models of similar capabilities.

Examining the archive generated by Llama-3-70B with ACES-ELM, Mistral Large 2 achieved a 70% pass@k (74% for Llama-3-405B), while Llama-3-70B has a pass@1 of 36.8%. This demonstrates Mistral-Large and Llama-405B's superior performances as solvers. However, even with these state-of-the-art models, the benchmark generated by Llama-3-70B is not saturated, as there is still approximately 30% room for improvement to fully solve it.

### New finetuning results
To address concerns of Reviewers ESVE and NuHw about applications and model improvement, we performed experiments with finetuning. We finetuned the Llama-3-8b model using datasets generated by WizardCoder (variant of the state of the art method WizardLM for generating synthetic data [4]), StaticGen (established baselines), and our proposed ACES-ELM method.  We then evaluated the model's performance using the greedy pass@1 metric on a series of test sets. These test sets were equally composed of puzzles from our method and StaticGen with increasing difficulty levels (k value in the Figure 1.d), generated using a different seed than the training data.

The results of this experiment, illustrated in Figure 1.d, reveal several findings that highlight the superiority of ACES-ELM. On the most challenging test set, the Llama-3-8b model finetuned with data generated by ACES-ELM achieved a remarkable pass@1 score of 53.3, significantly surpassing both baseline methods and also outperforming the Llama-3-70B model. This demonstrates the effectiveness of our ACES-ELM method in generating high-quality training data that enables models to tackle more complex coding tasks.
While the Llama-3-8b models fine-tuned with WizardCoder and StaticGen showed improvements over the baseline Llama-3-70B model (achieving pass@1 scores of 49.4 and 41.6, respectively), they consistently underperformed compared to the ACES-ELM-trained model. This underscores the superior quality of the training data generated by ACES-ELM. Moreover, as evident from Figure 1.d, the performance gap between ACES-ELM and the baseline methods becomes more pronounced as the difficulty level of the testset increases. This suggests that ACES-ELM is particularly effective in preparing models for more complex coding challenges.

---

We think we have clarified all the points the reviewers have raised and responded to concerns regarding the contribution, applications and generality of our method. We thus kindly ask reviewers to raise their grade if they feel their concerns have been addressed or otherwise provide detailed requirements that would convince them to do so.

[1] Evolution Through Large Models, Lehman et. al. 2023

[2] Quality-Diversity Through AI Feedback, Bradley et. al. 2023

[3] Rainbow Teaming: Open-Ended Generation of Diverse Adversarial Prompts, Samvelyan et. al. 2024

[4] WizardCoder: Empowering Code Large Language Models with Evol-Instruct, Luo et. al. 2023

---

### Decision · Program_Chairs · 2024-09-25

**Decision:**

Accept (spotlight)

**Comment:**

This paper presents a new method for automatic generation of challenging and diverse coding puzzles, which could be an important step towards open-ended improvement of LLMs. After discussion, 3/4 reviewers have voted for acceptance, with one reviewer strongly opposing acceptance. The main arguments given in favor of acceptance are that the method is interesting and novel and has high potential for impact. Reviewers also noted the solid validation and clear presentation. The authors have engaged thoroughly with all reviewers and addressed most concerns.

One reviewer (NuHw) remains skeptical, since the work relies too much on LLMs (which are unreliable) and follows a questionable approach (since the generated test functions could be wrong). This reviewer is generally skeptical about work that builds on LLMs, and believes that evaluation of code based on testing is inherently flawed, and formal verification is required.

In my opinion, the most important criticism is the validity of the generated test functions. Here I think the authors have done a good job addressing this concern. Specifically, if we take the generated test function f to *define* the problem, there is no way in which it could be invalid. By definition, a function g solves the problem if f(g()) == True. The question of whether f corresponds to some text description is secondary, since the text does not formally define the problem, and need not be passed to the LLM that is being evaluated. Thus I do not agree that the method is dubious.

Given that the majority of reviewers agree the paper is worth accepting, and the rebuttal of all significant criticisms by the authors, I recommend for this paper to be accepted at NeurIPS.